# Evaluation of global simulations of aerosol particle and cloud condensation nuclei number, and implications for cloud droplet formation

George S. Fanourgakis[1], Maria Kanakidou[1], Athanasios Nenes[2,3,4], Susanne E. Bauer[5,6], Tommi Bergman[7], Ken S. Carslaw[8], Alf Grini[9], Douglas S. Hamilton[10], Jill S. Johnson[8], Vlassis A. Karydis[11,12], Alf Kirkevåg[13], John K. Kodros[14], Ulrike Lohmann[15], Gan Luo[16], Risto Makkonen[17,18], Hitoshi Matsui[19], David Neubauer[15], Jeffrey R. Pierce [14], Julia Schmale [20], Philip Stier[21], Kostas Tsigaridis[6,5], Twan van Noije[7], Hailong Wang[22], Duncan Watson-Parris[21], Daniel M. Westervelt[23,5], Yang Yang[22], Masaru Yoshioka[8], Nikos Daskalakis[24], Stefano Decesari[25], Martin Gysel-Beer[20], Nikos Kalivitis[1], Xiaohong Liu[26], Natalie M. Mahowald[10], Stelios Myriokefalitakis[27], Roland Schrödner[28], Maria Sfakianaki[1], Alexandra P. Tsimpidi[11], Mingxuan Wu[26], Fangqun Yu[16]

[1]Environmental Chemical Processes Laboratory, Department of Chemistry, University of Crete, Heraklion, 70013, Greece

[2]Laboratory of Atmospheric Processes and their Impacts, School of Architecture, Civil & Environmental Engineering, Ecole Polytechnique Federale de Lausanne, Lausanne, 1015, Switzerland

[3]IERSD, National Observatory of Athens, P. Penteli 15236, Athens, Greece

[4] ICE-HT, Foundation for Research and Technology – Hellas, Greece

[5] NASA Goddard Institute for Space Studies, New York NY USA

[6] Center for Climate Systems Research, Columbia University, New York NY USA

[7] Royal Netherlands Meteorological Institute (KNMI), De Bilt, Netherlands

[8] School of Earth and Environment, University of Leeds, UK

[9] independent researcher

[10] Department of Earth and Atmospheric Sciences, Atkinson Center for a Sustainable Future, Cornell University, Ithaca, NY, USA.

[11] Department of Atmospheric Chemistry, Max Planck Institute for Chemistry, Mainz, German

[12]Forschungszentrum Jülich, Inst Energy & Climate Res IEK-8, D-52425 Jülich, Germany

[13] Norwegian Meteorological Institute, Oslo, Norway

[14] Department of Atmospheric Science, Colorado State University, Fort Collins, Colorado, USA

[15] Institute for Atmospheric and Climate Science, ETH Zurich, Zurich, Switzerland

[16] The Atmospheric Sciences Research Center (ASRC), of the State University of New York at Albany

[17] System Research, Finnish Meteorological Institute, P.O. Box 503, 00101, Helsinki, Finland

[18] Institute for Atmospheric and Earth System Research / Physics, University of Helsinki, P.O. Box 64, 00014 Helsinki, Finland

[19] Graduate School of Environmental Studies, Nagoya University, Nagoya, Japan.

[20] Laboratory of Atmospheric Chemistry, Paul Scherrer Institute, Villigen, Switzerland

[21] Atmospheric, Oceanic & Planetary Physics, Department of Physics, University of Oxford, UK

[22] Atmospheric Sciences and Global Change Division, Pacific Northwest National Laboratory, Richland, Washington, USA

[23] Lamont-Doherty Earth Observatory, Columbia University, Palisades, NY, USA, 10964

[24] Laboratory for Modeling and Observation of the Earth System (LAMOS) Institute of Environmental Physics (IUP), University of Bremen, Bremen, Germany

[25] Institute of Atmospheric Sciences and Climate, National Research Council of Italy, Via Piero Gobetti, 101, 40129 Bolonga, Italy

[26] Department of Atmospheric Science, University of Wyoming, Laramie, Wyoming, USA

[27] Institute for Environmental Research and Sustainable Development, National Observatory of Athens, Penteli, Greece

[28] Centre for Environmental and Climate Research, Lund University, Sweden

*Correspondence to*: Maria Kanakidou (mariak@uoc.gr), Athanasios Nenes (athanasios.nenes@epfl.ch )

**Abstract.** A total of sixteen global chemistry transport models and general circulation models have participated in this study. Fourteen models have been evaluated with regard to their ability to reproduce near-surface observed number concentration of aerosol particles and cloud condensation nuclei (CCN), and derived cloud droplet number concentration (CDNC). Model results for the period 2011-2015 are compared with aerosol measurements (aerosol particle number, CCN and aerosol

particle composition in the submicron fraction) from nine surface stations, located in Europe and Japan. The evaluation focuses on the ability of models to simulate the average across time state in diverse environments, and on the seasonal and short-term variability in the aerosol properties.

There is no single model that systematically performs best across all environments represented by the observations. Models tend to underestimate the observed aerosol particle and CCN number concentrations, with average normalized mean

bias (NMB) of all models and for all stations, where data are available, of -24% and -35% for particles with dry diameters > 50nm and >120 nm, and -36% and -34% for CCN at supersaturations of 0.2% and 1.0%, respectively. However, they seem to behave differently for particles activating at very low supersaturations (<0.1%) than at higher ones. Fifteen models have been used to produce ensemble annual median distributions of relevant parameters. The model diversity (defined as the ratio of standard deviation to mean) is up to about 3 for simulated $N_3$ (number concentration of particles with dry diameters larger

than 3 nm) and up to about 1 for simulated CCN in the extra-polar regions. A global mean reduction of a factor of about 2 is found in the model diversity for $CCN_{0.2}$ compared to that for $N_3$, maximizing over regions where new particle formation is important.

An additional model has been used to investigate potential causes of model diversity in CCN and bias compared to the observations by performing a perturbed parameter ensemble (PPE) accounting for uncertainties in 26 aerosol-related model

input parameters. This PPE suggests that biogenic secondary organic aerosol formation and the hygroscopic properties of the organic material are likely to be the major sources of CCN uncertainty in summer, with dry deposition and cloud processing being dominant in winter.

Models capture the relative amplitude of seasonal variability of the aerosol particle number concentration for all studied particle sizes with available observations (dry diameters larger than 50, 80 and 120 nm). The short-term persistence time (on

the order of a few days) of CCN concentrations, which is a measure of aerosol dynamic behavior in the models, is underestimated on average by the models by 40% during winter and 20% in summer.

In contrast to the large spread in simulated aerosol particle and CCN number concentrations, the CDNC derived from simulated CCN spectra is less diverse and in better agreement with CDNC estimates consistently derived from the observations (average NMB -13% and -22% for updraft velocities 0.3 and 0.6 m.s$^{-1}$, respectively). In addition, simulated

CDNC is in slightly better agreement with observationally-derived value at lower than at higher updraft velocities (index-of-agreement of 0.64 vs 0.65). The reduced spread of CDNC compared to that of CCN is attributed to the sublinear response of CDNC to aerosol particle number variations and the negative correlation between the sensitivities of CDNC to aerosol particle number concentration ($\partial Nd/\partial Na$) and to updraft velocity ($\partial Nd/\partial w$). Overall, we find that while CCN is controlled by both aerosol particle number and composition, CDNC is sensitive to CCN at low and moderate CCN concentrations and

to the updraft velocity when CCN levels are high. Discrepancies are found in sensitivities $\partial Nd/\partial Na$ and $\partial Nd/\partial w$; models may be predisposed to be too "aerosol-sensitive" or "aerosol-insensitive" in aerosol-cloud-climate interaction studies, even if

they may capture average droplet numbers well. This is a subtle, but profound finding that only the sensitivities can clearly reveal and may explain inter-model biases on the aerosol indirect effect.

## 1 Introduction

Aerosol particles absorb and scatter radiation, thereby modulating the planetary radiative balance (Boucher et al., 2013; Myhre et al., 2013). They also provide the nuclei upon which cloud droplets and ice crystals form; variations thereof can profoundly impact cloud formation and precipitation. Both the direct radiative effects of aerosols and their impacts on clouds are thought to be important for climate at global and regional scales, although they are highly uncertain and confound projections of anthropogenic climate change (e.g., Boucher et al., 2013; Seinfeld et al., 2016). The impacts of aerosols on clouds in particular introduce considerable uncertainty in our estimates of equilibrium climate sensitivity and transient climate response to the combined changes in aerosol and greenhouse gases concentrations (e.g., Seinfeld et al., 2016; Fan et al. 2016).

Aerosols can be either directly emitted from a variety of sources (primary aerosols) or formed by nucleation from precursor compounds (secondary aerosols), which afterwards can grow by condensation and coagulation from a few nanometers to a few hundreds of nanometers (Kerminen et al., 2012). Note that secondary aerosol also includes the condensed material upon primary emitted aerosol. Aerosols that have the potential to create cloud droplets at atmospherically-relevant conditions are termed cloud condensation nuclei (CCN). The CCN number concentration depends on the particle size distribution, chemical composition and mixing state, as well as the level of water vapor supersaturation that develops in rising air parcels (Köhler, 1936; Seinfeld and Pandis, 2006). It is now established that primary emissions of particulate matter and particle formation from anthropogenic precursor gases have strongly modulated clouds and climate at the global scale since the industrial revolution (Boucher et al., 2013). Much work remains, however, to reduce the uncertainty associated with anthropogenic aerosol-cloud-climate interactions.

Among the main sources of uncertainty in simulating aerosol microphysics at regional to global scales are the amounts of particle and precursor vapor mass emitted by anthropogenic activities or natural sources, as well as the size distribution of the emitted particles and their representation in models. However, Mann et al. (2012) showed that a careful choice of the aerosol parameters describing the aerosol distribution can reduce differences between the sectional and the modal description of aerosol microphysics in most parts of the atmosphere. Furthermore, carbonaceous combustion aerosol, although assumed hydrophobic upon emission, was found to contribute up to 64% of global surface CCN concentrations (Spracklen et al., 2011). Although less important than particle size for CCN formation, particle chemical composition determines aerosol hygroscopicity (Twomey, 1977; Dusek et al., 2006; Petters and Kreidenweis, 2007; Cubison et al., 2008; Bougiatioti et al., 2017). Adequate description of aerosol hygroscopicity is required to accurately describe CCN and cloud droplet number variability. In this respect, uncertainties are partially related to the organic aerosol (OA), which can be composed of thousands compounds with different physical and chemical properties. OA contributes to the fine aerosol mass by up to 30-70% depending on location and season (Kanakidou et al., 2005; Jimenez et al., 2009); while source estimates of OA are spanning one order of magnitude (see the AeroCom phase-II intercomparison study of 31 models by Tsigaridis et al. (2014)). Regionally, sea salt (SS) and mineral dust (DU) are also significant contributors to the total aerosol particle mass and number concentration. Atmospheric mass loads during the first phase of AeroCom showed a high diversity among 15 models of 54% for SS and 40% for DU (Textor et al., 2006). This diversity arises from the different parameterizations used to calculate the size-resolved fluxes and their dependence on wind speed but also from the consideration, or not, of the super coarse aerosol fraction (Huneeus et al., 2011; Tsigaridis et al., 2013). Although nitrate ($NO_3^-$) and ammonium ($NH_4^+$) are not explicitly studied here, differences up to a factor of 13 in the atmospheric burden of $NO_3^-$ and 17 and 4 for $NH_3$ and $NH_4^+$, respectively, have been found between AEROCOM models (Bian et al., 2017).

Formation of new particles by nucleation in the atmosphere is a frequent phenomenon in the free troposphere and in the continental boundary layer (e.g. Kerminen et al., 2010; Kulmala and Kerminen, 2008) and an important source of aerosol particle number on a global scale (Kerminen et al., 2012; Kalivitis et al., 2015; Gordon et al., 2017). Although it is well established that sulfuric acid, due to its low volatility, plays a central role in new particle formation and growth, it cannot

explain the observed substantial growth of small particles in many environments where organics and $NH_3$ are abundant. This is due to the low concentration of sulfuric acid and is evidenced by the observed poor correlation of its concentration with the very small particles (e.g. Pierce et al., 2011). Recently, the involvement of organics from early stages of nucleation and growth of particles has been established (e.g. D'Andrea et al., 2013; Spracklen et al., 2008; Makkonen et al., 2009; Tröstl et al., 2016). Several approaches for modeling particle growth in large-scale models have been developed, which are very

sensitive to the volatility of organic vapors (e.g. Laaksonen et al., 2008; Yu, 2011; D'Andrea et al., 2013) and are being implemented in global models.

The number concentration and the size of cloud droplets depend both on the concentrations of CCN and on the cloud updraft velocity (Pruppacher and Klett, 1997; Seinfeld and Pandis, 2016). However, the spatial scale of updrafts governing droplet formation is several orders of magnitude smaller than the size of the grid boxes of global models. Therefore,

parameterized aerosol-cloud interactions in climate models require sub-grid scale vertical velocity distributions to calculate grid-scale relevant cloud droplet number concentration (CDNC) (Morales et al., 2010). Karydis et al. (2012) and Moore et al. (2013) have shown that in regions with low particle number concentrations, such as the Arctic and remote oceans, CDNC is more sensitive to CCN uncertainty than in continental regions where particle number concentrations exceed $10^4$ $cm^{-3}$. In contrast, Ervens et al. (2010) pointed out that at high updraft velocities, supersaturation is controlled by adiabatic cooling,

and CDNC is not very sensitive to errors in simulated CCN number concentration. They estimated that uncertainties in the chemical composition of aerosol particles that could lead to a doubling of CCN concentration, would affect CDNC by only about 10-20%. Therefore, there are two distinct regimes with regard to CDNC sensitivity: the aerosol-limited and the updraft velocity-limited ones (Reutter et al., 2009).

Totally different cloud radiative (indirect) effects could be computed by climate models depending on the dominance of

CDNC sensitivity to either aerosol number or updraft velocity (Sullivan et al., 2016). Therefore, capturing the balance between the two is critical in understanding where and when aerosol emissions are governing the variability of cloud properties and where the updraft velocity is the controlling factor. Failure of state-of-the-art models to capture such sensitivity implies that even if models exhibit similar magnitude of aerosol indirect effects, it may be for completely different reasons (Sullivan et al., 2016). In this case models would show limited skills and their predictions would be

associated with low confidence.

The aims of this work are to *i*) assess the accuracy of state-of-the-art global aerosol models in simulating the chemical composition and number concentration of aerosol particles, with focus on CCN concentrations at various water vapor supersaturation ratios, *ii*) document the diversity of the global models in simulating these aerosol properties, *iii*) produce an ensemble view of the global distribution of aerosol particle and CCN number concentrations, together with the most

important particle chemical components at the Earth's surface, *iv*) evaluate the agreement of inferred CDNC from modeled and from observed CCN spectra and their sensitivity to aerosol number concentrations and updraft velocities, *v*) evaluate the potential causes of model diversity and bias versus observations using model uncertainty analysis, and, *vi)* provide recommendations for future model improvements.

Sixteen global models contributed to this study, and multi-year observations of CCN, size-resolved particle number

concentration distributions, and particle chemical composition obtained from eight atmospheric monitoring stations in Europe and one in Japan were used as observational reference, representing distinct atmospheric environments (Schmale et al., 2017, 2018).

## 2 Methodology

### 2.1 Contributing models and model description

Model set-up, such as spatial resolution, meteorological conditions and emission inventories differ significantly among models (supplementary Tables S1 to S4). The spatial resolution varies among the models from 0.94° by 1.3° to 4° by 5.0° (latitude by longitude) and from 25 to 56 vertical layers up to 10 and even 0.1 hPa. Nine of the models are general circulation models (GCMs) and six are chemical transport models (CTMs). The CTMs use prescribed (and different) meteorological data sets; while the GCMs (with the exception of GISS-E2-TOMAS) are nudged to various reanalysis products. Atmospheric transport, secondary aerosol formation and removal of aerosols are driven by wind, temperature, radiation, precipitation and relative humidity, as well as cloud fraction and liquid water content. In addition, most of the models use wind-driven dust, sea salt, and marine organic aerosol emissions as well as calculated online biogenic emissions of non-methane volatile organic compounds (NMVOC) (Table S3). Therefore, meteorology significantly affects number concentration, composition and other metrics of aerosol particles.

Despite the recognized importance of organic compounds in nucleation (Tröstl et al., 2016), several global models that participated in the present study use binary homogeneous nucleation of sulfuric acid and water (referred later as BHN e.g. Kulmala et al. (1998), Vehkamäki (2002)) and contribution of organics to particle growth (see supplementary section S1 and Table S2 and references therein). GEOS-chem-TOMAS assumes ternary nucleation mechanism when $NH_3$ is present and a binary one when $NH_3$ is absent. GEOS-Chem-APM and CAM5-Chem-APM employ ternary ion-mediated nucleation (TIMN) scheme which considers both binary and ternary as well as ion-mediated and neutral nucleation (Yu et al., 2018). New particle formation in TM5 is calculated as combination of BHN and organic-sulfuric acid nucleation (Riccobono et al., 2014).

Once in the atmosphere, aerosols undergo transformations through chemical and physical processes, such as coagulation, condensation and evaporation that modify their size and physical and chemical properties. These aerosol microphysical processes are parameterized differently in models. Eight of the models use modal schemes in which the evolution of particle number and mass concentrations are described by log-normal distributions, and the remaining models use the sectional approach with various numbers of monodisperse size-bins describing aerosol particle number concentration and chemical composition (Table S2).

Regarding the eight modal models, six of them (the three ECHAM models, EMAC, TM4-ECPL and TM5) are based on the M7 aerosol module developed by Vignati et al. (2004) for the description of aerosol microphysics, or improved versions of M7 to account for $SO_2$ oxidation to sulfuric acid, contribution of organics to growth, and additional aerosol species. Other aerosol microphysics modules used in models participating in this study are the Modal Aerosol Modules (MAM3 and MAM4; (Liu et al., 2012; Liu et al., 2016), the Advanced Particle Microphysics (APM) package (Yu and Luo, 2009; Yu, 2011; Yu et al., 2018), the TwO-Moment Aerosol Sectional (TOMAS) microphysics package (Adams and Seinfeld, 2002), the Multiconfiguration Aerosol Tracker of mIXing state (MATRIX) module (e.g. Bauer et al., 2008), the Aerosol Two-dimensional bin module for formation and Aging Simulation version 2 (ATRAS2; Matsui, 2017) and a production tagged module OsloAero5.3 used in combination with the offline microphysics scheme AeroTab5.3 (Kirkevåg et al., 2018). Supplementary tables S1, S2, S3, and S4 provide a summary of the main features of the participating models and appropriate references.

Relevant to this study are also differences in the aerosol components that are taken into consideration in the models for the CCN calculations. Nine models (CAM5- MAM3, CAM5-MAM4, CAM5.3-Oslo, the three ECHAM models, GEOS-Chem-TOMAS, GISS-E2-TOMAS models and TM4-ECPL) do not account for particulate nitrate at all or in the CCN calculations (supplementary Table S2). TM4-ECPL however computes $NO_3^-$ and $NH_4^+$ mass distribution in fine and coarse modes by the ISORROPIA II module (Fountoukis and Nenes, 2007). Similarly TM5 uses EQSAM (Metzger et al., 2002b,

2002a) to calculate, using bulk aerosol approach, the partitioning of ammonium nitrate between gaseous and particulate phase with the particulate mass assumed to reside in soluble accumulation mode.

Both dry deposition and wet deposition of aerosol particles are taken into account in the participating models as shown in the supplementary Table S4. For the dry deposition models account for gravitational settling and for turbulence, thus these processes depend on the aerosol particle size. The omission of super-coarse particle sources associated with dust and sea-salt particles result in discrepancies between models, and, between model results and observations (Myriokefalitakis et al., 2016). Wet deposition parameterizations account for both in-cloud scavenging, which is sensitive to the solubility of aerosol particles, and below-cloud scavenging by convective and large-scale precipitation (Seinfeld and Pandis, 2006). In addition, while all models account for in-cloud scavenging of aerosols and for the aerosol release from evaporation of droplets, a few models account also for melting and sublimation of ice crystals. For the calculation of CCN concentrations from the aerosol number and mass distributions, models need to specify their hygroscopicity from the volume-weighted hygroscopicities of each constituent (Table 1) following the approach of Petters and Kreidenweis (2007).

Furthermore, most of the participating models (supplementary Table S4) follow the AEROCOM recommendation of biomass burning emission heights which in the boreal regions extend above 2 km and up to 6 km for the Canadian boreal fires (Dentener et al., 2006). ECHAM6-HAM2 and ECHAM6-HAM2-AP use a slightly different vertical distribution of biomass burning emissions with 75% within the planetary boundary layer (PBL), 17% in the first and 8% in the second level above the PBL (Tegen et al., 2019). EMAC assumes biomass burning emissions at 140 m and GEOS-Chem-APM well mixed in the boundary layer.

In addition to these 15 models, we include the results from perturbed parameter ensemble (PPE) simulations using HadGEM3-UKCA (Yoshioka et al., 2019; see details in supplementary section S1). The PPE consists of 235 atmosphere-only simulations for the year 2008 with 26 parameters controlling aerosol emissions and processes perturbed simultaneously. Simulations were nudged to ERA-Interim wind and temperature and all aerosol feedbacks to atmospheric dynamics are turned off. Therefore, all simulations share the same meteorology. CCN number concentrations were calculated globally for all member simulations and taken at geographical locations and elevations of observation stations. These simulations were then used to create Gaussian process emulators at each station location from which 260,000 'model variants' were generated that densely sample the 26-dimension parameter space. The emulators were validated against additional model simulations to show that the emulator uncertainty is much smaller than the model parametric uncertainty.

## 2.2 Observational data for model evaluation

Datasets of CCN at various supersaturations, particle number concentrations, size distributions and particle chemical composition measured at one atmospheric monitoring station in Japan and eight Aerosols, Clouds, and Trace gases Research InfraStructure (ACTRIS) atmospheric monitoring stations in Europe (Schmale et al., 2017) were used in the present study (Figure 1) for evaluation of model results. The observatories are representative of different environments (Pacific, Atlantic and Mediterranean marine atmospheres, high alpine and boreal forest continental atmospheres). A brief site description of the observatories is provided in the supplementary Table S5, while more technical details are given by Schmale et al. (2017). While in general measurement data are available from the period of 2011 to 2015, each station covered only a sub-period of those five years, but at least one entire year (Schmale et al., 2017). Despite using point measurements, the long period of observations allows evaluation of the global models without biases associated with the model resolution (Schutgens et al., 2016). Six out of the nine stations provided non-refractory chemical composition data of submicron particles (based on aerosol mass spectrometry); while all stations recorded submicron particle number size distributions and CCN number concentrations over a variety of supersaturations. A detailed discussion of the observational results can be found in Schmale et al. (2018).

For this study, the observations of CCN concentrations at supersaturations spanning between 0.1 and 1.0%, the number concentrations of aerosols with dry diameters larger than 50, 80, and 120 nm (denoted hereafter as $N_{50}$, $N_{80}$, $N_{120}$, respectively), as well as $PM_1$ (particles with dry diameters less than 1 μm) chemical composition (mainly sulphate ($SO_4^{2-}$ hereafter SO4), organic aerosol (OA)) from the nine stations are used. The CCN data for these stations cover at least 75% of each year (Schmale et al., 2017). Observational data have been further filtered so that there is a minimum data requirement, which means that daily averages are calculated from hourly data only for days with at least six hourly measurements. Monthly averages follow similar method, where the average is calculated only for months with at least 10 daily averages. When fewer data are available, the data are not considered representative of this quantity and are not included in the comparisons with the model results.

## 2.3 Design of the experiment

This model experiment has been designed within the BACCHUS EU project and has been opened for participation to the entire AEROCOM global modeling community. Global simulations have been performed for the years 2010-2015 (2010 is used as a spin-up). SO4, BC, OA, SS and DU are the aerosol components that are considered here. Models provided hourly values for the $N_{50}$, $N_{80}$, $N_{120}$; CCN number concentrations for thirteen supersaturations ranging from 0.05% up to 1.0% (that are 0.05%, 0.075%, 0.1%, 0.15% and from 0.2% to 1.0% in increments of 0.1%, denoted hereafter as $CCN_i$, where i=supersaturation value) as well as the chemical composition of $PM_1$ particles at the station locations (Table S5). The large number of different supersaturations at which CCN are computed allows for direct comparisons with all available observations of CCN for the nine stations as well as for the calculation of CDNC (Section 2.4). Among the models that participated in the present study GISS-E2-TOMAS and HadGEM3-UKCA did not provide any results for the stations due to meteorology not corresponding to the measurement time period (free-running for the first one and 2008 for the second); therefore, all multi-model median (MMM) for the stations presented below have been computed excluding these models.

Beyond station data, the global annual mean surface distribution of the $CCN_{0.2}$, the particle numbers $N_3$, $N_{50}$ and $N_{120}$ and the mass composition of the $PM_1$ particles for the year 2011 are provided by fifteen models (HadGEM3-UKCA did not provide such results). The MMM has been computed as the median of the contributing models.

In addition to the data provided by the 15 global models, the results of the PPE using HadGEM3-UKCA (Yoshika et al, 2019) are used in this study to quantify the model parametric uncertainty in CCN and to perform a sensitivity analysis to quantify how each parameter contributes to the overall uncertainty.

## 2.4 Data interpretation methodology

*CCN Persistence.* To investigate the duration for which the CCN number concentration remains similar to its earlier concentration, the so-called persistence, the autocorrelation function (ACF) of the CCN time series has been calculated as in Schmale et al. (2018) (see also supplementary S2). This ACF may provide valuable information about the drivers of the variability of the CCN number concentration in the atmosphere. In the present study, we chose to compute the ACF based on model results of $CCN_{0.2}$ at the 9 sampling sites and compare them with the corresponding ACF obtained from observations (Schmale et al., 2018). For a direct comparison, we use the same time periods as for the observations, which vary among the sampling sites. For all ACF calculations, hourly data of $CCN_{0.2}$ were used, for both the observations and model results.

*CDNC calculations.* While GCMs calculate CDNC using a variety of approaches, for the present study CDNC is calculated off-line, using a common parameterization for CCN spectra derived from the models or from the observations. This approach allows understanding the importance of differences in modeled and observed CCN spectra by expressing them as differences in CDNC that would form in a given type of cloud. We have calculated CDNC for two different updraft velocities: one characteristic for stratiform clouds ($w = 0.3 \, \mathrm{ms}^{-1}$) and the second characteristic for cumulus clouds ($w =$

0.6 ms$^{-1}$). Similar calculations have been performed using the observed CCN spectra at the stations, where such information is available, to enable comparison of model results with observations. The ability of the modeled CCN spectra to reproduce the observed sensitivity of CDNC to aerosol or to updraft velocity is also evaluated. Note that evaluation of the differences in CDNC calculations by the different models that are derived both from the parameterizations used and from their input variables would require a different model intercomparison design than here and is planned for the future. Morales-Betancourt et al. (2014a) provide a good example, where the source of CDNC prediction discrepancy for two state-of-the-art parameterizations in the CAM5 global model was unravelled using adjoint sensitivity analysis. That study pointed to exactly which aspects of the parameterization (i.e., water uptake from large CCN) were not captured adequately, leading to the highly improved droplet parameterizations (Morales-Betancourt and Nenes, 2014b) that was used in the current study.

The calculation of CDNC is based on the parameterization of Nenes and Seinfeld (2003) with the mass transfer augmentations proposed by Fountoukis and Nenes (2005), Barahona and Nenes (2007) and Morales Betancourt and Nenes (2014). Using the CCN at different supersaturations (section 2.3) allows to consistently construct the CCN spectrum function $F(s)$ from each simulation - which provides the CCN number as a function of supersaturation, $s$ (Sotiropoulou et al., 2006):

$$F(s) = \frac{N}{1 + \left(\frac{s}{b}\right)^a} \qquad (2)$$

where $N$ is the total number of particles, and $a, b$ are parameters determined using a non-linear fitting procedure for each one of the participating models. $F(s)$ is then computed for each station's grid point and time step of the model outputs (with $b$ and $a$ being fitting parameters), and CDNC, denoted in the figures by $N_d$, is computed from the parameterization for prescribed values of the vertical velocity. This fitting approach has been also applied to the CCN observations since they are available only for a limited number of supersaturations; and thus cannot be directly used for accurate calculation of CDNC. A well-constrained CCN spectrum requires concentrations for at least five different supersaturations at the same time instance (Sotiropoulou et al., 2006). Such information was available only at five stations (Cabauw, Finokalia, Jungfraujoch, Mace Head and Vavihill), which are subsequently used for deriving CDNC based on observations and compared against model-derived CDNC.

The CDNC parameterization uses as input $F(s)$, cloud-base pressure and temperature, and the vertical velocity characterizing the cloud updraft (either as a single updraft, or a "characteristic" value that provides a distribution-averaged value; Morales and Nenes, 2010). It determines the value of maximum supersaturation, $s_{max}$, that develops in the cloudy updrafts, using the concept of "Population Splitting" (Nenes and Seinfeld, 2003). $s_{max}$ is achieved during the cloud parcel ascent and is calculated considering the water vapor balance between its availability from cooling and its loss from condensational growth of the CCN (Fountoukis and Nenes, 2005). CDNC is then obtained from the CCN spectrum as $N_d=F(s_{max})$. This approach works well for stratus and stratocumulus clouds (Morales and Nenes, 2010). CDNC calculated here is from primary activation and does not consider the influence of pre-existing droplets, although modifications to the parameterization can account for this as well (e.g., Barahona et al., 2014).

***Ensemble modeling computation.*** The modeled hourly aerosol particle number concentrations, mass composition, CCN and CDNC at the nine stations have been used to calculate daily and monthly averages. Comparison of individual model results with observations is provided in the Supplementary Material Figures S2 and S3, because it can be used to identify strengths and weaknesses of each specific model and can serve as a guide for model improvements in the future. In Section 3, the multi-model median (MMM) is compared to observations. The diversity of the model results (defined as the ratio of standard deviation-to-mean) and the mean of the models, which in several cases significantly differs from the MMM, are also reported in these comparisons.

Annual averages of the global surface distributions of $N_3$, $N_{50}$, $N_{120}$, $CCN_{0.2}$ and $PM_1$ mass concentrations of the major aerosol components have been provided by a total of fifteen models. Global fields have first been re-gridded to a 5°×5° grid for all models, which is close to the coarsest-resolved participating models (4°x5°). Then the MMM and diversity are

calculated, as described above, for the stations. Note that 5°×5° is a very coarse grid size, which no doubt affects the model-to-observations comparison, particularly when comparing to sites within small heavily polluted area where a large rural background is now also being added in and vice versa. Therefore, it is worth mentioning that the surface stations used for model comparison are representative of the larger area in which they are located and justify our choice for a relatively large grid to re-grid all model results. For the mountain stations, the appropriate model level has been considered that corresponds to the station's altitude above sea level. Annual means of the individual models are also presented in the Supplementary Material (Figures S6 to S14).

***Performance indexes.*** For the comparison of model results with observations, a number of statistics variables have been calculated and defined as shown in the supplementary material S3.2. Hereafter we discuss

$$\text{the Index-of-Agreement (IOA} = 1 - \frac{\sum_{i=1}^{N}(P_i - O_i)^2}{\sum_{i=1}^{N}(|O_i - \bar{O}| + |P_i - \bar{O}|)^2}),$$

$$\text{the normalized mean bias (NMB} = \frac{\sum_{i=1}^{N}(P_i - O_i)}{\sum_{i=1}^{N} O_i} \times 100\%)$$

$$\text{and the normalized mean error (NME} = \frac{\sum_{i=1}^{N}|P_i - O_i|}{\sum_{i=1}^{N} O_i} \times 100\%),$$

where M are model results, O are observations and NMB, NME and IoA are used to quantify the performance of the models to reproduce observations. IoA is a measure of the agreement of model results with the observations. In this study we use all three for the evaluation of the capability of the models to reproduce the observations. We calculate also

$$\text{the Pearson linear regression coefficient (} r = \left[\frac{\sum_{i=1}^{N}(P_i - \bar{P})(O_i - \bar{O})}{\sqrt{\sum_{i=1}^{N}(P_i - \bar{P})^2}\sqrt{\sum_{i=1}^{N}(O_i - \bar{O})^2}}\right])$$

as a measure of the ability of the model results to represent the variability in the observations.

## 3 Evaluation against station observations

### 3.1 CCN number concentrations comparisons with multi-model median

The models tend to underestimate the monthly $CCN_{0.2}$ number concentration in the lowest model level at all sites (Fig. 2 and supplementary Fig. S2) for the years 2011-2015: Average normalized mean bias (NMB) of all models and for the nine sites is as low as -36% and the normalized mean error (NME) is 69%; while among individual models and stations NMB and NME vary from -88% to 145% and from 40% to 159%, respectively (see supplementary section S3.2 for definitions and Table S6 for results). The Finokalia station is an exception, where most models overestimate $CCN_{0.2}$ (average NMB around 47%) with eight models showing significant overestimation (NMB>10%) and six models smaller deviations from observations (-10%<NMB<10%). Among the studied locations, Finokalia is the station with the highest observed critical diameter (~200 nm at a supersaturation of 0.2% according to Schmale et al., 2018), therefore, potential inaccuracies in the model-determination of the critical size may be responsible for the model overestimate of $CCN_{0.2}$ at this station.

Such a hypothesis is supported by earlier studies that have observed large size-dependence of the sensitivity in activation fraction at low supersaturations and in the size ranges between 60 and 100 nm (Bougiatioti et al., 2011). Deng et al. (2013) reported inferred critical diameters varying by factors of 2-3 for low supersaturations from 0.06% to 0.2% and suggested the use of size-resolved particle number concentrations with inferred critical diameters or size-resolved activation ratios to predict CCN. Errors in CCN predictions have been shown to exceed 50% only at very low supersaturations (Reutter et al., 2009) and reaching a factor of 2.4, while at high supersaturations CCN overestimate can be less than 5% (Ervens et al., 2007). The global near-surface mean CCN prediction error has been estimated at about 9% and regionally the maximum error can reach 40% (Sotiropoulou et al., 2007). The largest CCN prediction error was found in regions with low in-cloud

$s_{max}$, like those affected by long-range transport of pollution or industrial pollution plumes, and the lower CCN prediction error in regions where in-cloud $s_{max}$ is high, which is typical for pristine areas. Sotiropoulou et al. (2007) also found that the assumption of size-invariant chemical composition of internally mixed aerosol increases the error by a factor of two.

The underestimation of the observed $CCN_{0.2}$ by the models is largest at the high alpine site of Jungfraujoch (mean NMB of all models: -73%), where none of the models is able to capture the maximum observed values of $CCN_{0.2}$ (~300-600 cm$^{-3}$) during summer. Deficiencies in the models' representation of the boundary layer and mixing of air between the boundary layer and the free troposphere in complex terrain like the Alps as well as the sampling of the models based on the station's altitude might be reasons for this systematic underestimation by the models (D'Andrea et al., 2016). Despite the quantitative differences in the estimation of the $CCN_{0.2}$ concentrations, models are able to qualitatively capture the relative differences in $CCN_{0.2}$ concentrations between stations, as well as their seasonal variations. Comparing the $CCN_{0.2}$ as calculated from the observations and as computed from the daily MMM for the days with available observations for the stations, we find a Pearson linear correlation coefficient (r) that vary between 0.44 (for Melpitz) and 0.83 (for Mace Head), showing significant covariation of model results with observations. Furthermore, ranking the stations based on the observed mean $CCN_{0.2}$ levels (supplementary Figure S17) we find that the corresponding MMM mean follows this station ranking with the exception of Finokalia where, as further discussed, the models overestimate the observed $CCN_{0.2}$ although they capture well (r=0.76) the observed temporal variability. The MMM index-of-agreement (IoA) varies between 0.44 and 0.82 for the different stations with the best for Finokalia remote coastal station and the worst for Jungfraujoch alpine station. The largest difference in performance among models is found for the Mace Head station with an IoA varying between 0.20 and 0.89 for the individual models (Table S6).

To compare the calculated MMM and the observed seasonal variability of $CCN_{0.2}$ for each station (Fig. 3), the monthly model results have been temporally co-located with monthly mean observations. Furthermore, to increase clarity in Figure 3, for each station, the MMM $CCN_{0.2}$ has been multiplied by a scaling factor, $f$, so that the four season's mean of the simulated MMM $CCN_{0.2}$ concentrations becomes equal to the corresponding observed value. The factor $f$ is denoted for each station inside the frame. Overall, the seasonal pattern is nicely captured by the models, although the absolute values are underestimated everywhere (f >1.50) except at Finokalia (f=0.82) as discussed earlier.

For the high altitude continental background sites (Puy de Dôme, Jungfraujoch) low number concentrations with high seasonal variability are observed (winter (DJF) minimum and summer (JJA) maximum with observed ratios of summer-to-winter of 2.17 and 5.37, respectively, while the simulated MMM ratios are 3.19 and 5.58). This strong seasonality is attributed to changes in the height of the boundary layer that can affect these sites during summer but not during winter when the sites are mostly in the free troposphere (Schmale et al., 2018). At Jungfraujoch the boundary layer virtually never reaches up to the site. Instead, increased concentrations are caused by injections of boundary layer air into the lower free troposphere over the mountainous terrain. The free tropospheric background concentration of CCN is very low such that increases in number concentration of CCN-sized particles (90 nm in diameter) are a good indicator for boundary layer influence (Herrmann et al., 2015).

On the other hand, high $CCN_{0.2}$ number concentrations but low seasonal variability are found for the rural background stations of Cabauw and Melpitz, indicative of the elevated air pollution background in these regions. At these stations highest $CCN_{0.2}$ number concentrations are observed during spring, which are underestimated by the MMM. Furthermore, observations show a monotonous decrease from spring to summer and fall, while models calculated higher summertime values than in spring and fall at Cabauw and a monotonous increase from spring to fall at Melpitz. This could indicate that the models are not following the observed changes in the aerosol particle number concentration and/or the critical diameter at these stations (Schmale et al., 2018), possibly also associated with the adopted sizes in the primary aerosol emissions at these locations. At the other rural background station (Vavihill), both models and observations show lower $CCN_{0.2}$

concentrations and seasonal variability than at Cabauw or Melpitz. In addition, observations indicate a higher critical diameter at Vavihill (around 120 nm) than at the other two stations (around 90 nm) (Schmale et al., 2018).

Different seasonal cycles are also observed among the three coastal sites Mace Head, Finokalia and the Noto Peninsula: At the Mace Head site, due to the clean marine conditions over the Atlantic Ocean (Ovadnevaite et al., 2014), low $CCN_{0.2}$ concentrations are observed through the year. There, the highest concentrations are observed and simulated during spring. Both Finokalia and Noto Peninsula are impacted by long range transport that occurs through the free troposphere and affects the surface by mixing down into the boundary layer and the models qualitatively reproduce the observed seasonal cycles, simulating a high variation in the number concentration over the year. At Finokalia the observed and simulated summer seasonal maximum is also attributed to biomass burning plumes from north-east Europe (Bougiatioti et al., 2016), while high $CCN_{0.2}$ concentrations peaking in spring (observations available only for May) over the Noto Peninsula are due to pollutants originating from East Asia (Iwamoto et al., 2016; Schmale et al., 2018). However, the observed sharp decline of $CCN_{0.2}$ during the spring (May)-summer transition over the Noto Peninsula is also reproduced by the models. At Finokalia the models qualitatively follow the observed seasonality, although the observed summer-to-winter ratio (4.6) is underestimated by the models (2.3; Fig. 3). This can be due to the CCN sensitivity to loss by deposition during winter and to OA formation and hygroscopicity during summer that combined weaken the simulated seasonality (further discussion in section 5).

Finally, at Hyytiälä, on average the models calculate relatively small $CCN_{0.2}$ number concentrations and a low seasonal variability with a maximum in concentrations in summer, in agreement with observations, although they slightly underestimate the observed summer-to-winter ratio (1.5 modeled versus 1.7 observed). As discussed further in section 5, at Hyytiälä the modeled $CCN_{0.2}$ is very sensitive to errors in OA hygroscopicity and in secondary organic aerosol (SOA) formation from biogenic organic precursors during summer. Therefore, uncertainties in OA in the models and in particular underestimates of OA are expected to affect the summer-to-winter ratio.

Observed CCN number concentrations at the maximum supersaturation ratios measured at each station (which vary among stations, ranging from 0.7% to 1.0%) are compared to model results in Figure 4. CCN at various supersaturation ratios provides insights into the size distribution and the chemical composition in the models, since at high supersaturations smaller and less hygroscopic particles also activate. Most models underestimate CCN at high supersaturation at all stations with available observations (Figure 4), indicating that an insufficient number of small particles are predicted to activate in the model. However, observations are captured by the maximum and minimum of the 14 models (dashed green line) except for the alpine Jungfraujoch station. Overall, the average NMB and NME of all models and for all stations with available observations are -34% and 78% respectively; while among individual models and stations NMB varies from about -89% to about 253% (Table S6).

Comparing model performance for CCN at low supersaturation ($CCN_{0.2}$, Figure 2) and at high supersaturation ($CCN_{1.0}$, Figure 4), $CCN_{1.0}$ is systematically underestimated by the models across all stations. The NME of MMM for $CCN_{0.2}$ ranges from 45% (Finokalia) to 81% (Jungfraujoch) for the different stations with significant correlation coefficients between 0.44 (Melpitz) and 0.86 (Mace Head) indicating that the MMM model is able to simulate the temporal variability in the observations. For CCN at the highest supersaturation with available observations the NME varies from 50% (Finokalia) to 74% (Mace Head) and the correlation coefficients from 0.37 (Melpitz) to 0.78 (Mace Head) (see also supplementary Table S6). These results indicate that $CCN_{0.2}$ is in general better captured than CCN at higher supersaturations, both in absolute values and in temporal variability. Since the number concentration of CCN depends on both the chemical composition and the number of aerosol particles, it is worth investigating the role of these two factors separately.

### 3.2 CCN number concentrations comparisons with PPE

$CCN_{0.2}$ concentrations in perturbed parameter ensemble (PPE) simulations using HadGEM3-UKCA (Yoshioka et al., 2019) for 2008 at these stations are shown in Figure 5, together with observations. The blue solid line shows the emulator mean, the blue shading the range of one standard deviation around the mean, and the dotted lines the minimum and maximum emulator values. The range of one standard deviation either side of the mean value represents approximately 68% of all samples, therefore the blue shading shows approximately the same relative range as for the multi-model comparison in Figure 2 (25%/75% quartiles). The MMM averaged for the years 2011-2015 is also plotted in this figure for comparison purposes together with the 25%/75% quadrille shaded area. The mean of the available observations from the different years are shown by symbols. Since the interannual variability of simulated MMM $CCN_{0.2}$ concentrations shown in Figure 2 is generally small compared to inter-model variability, the difference in years between simulations and observations is not considered to undermine the model-data comparisons.

Except for Mace Head, the uncertainty ranges in the PPE are somewhat smaller than the 25%/75% quartiles of the models shown in Figure 2. This suggests model structural differences and emission inventories used in different models are more important source of diversity of estimated $CCN_{0.2}$ concentrations for central 70% range than fully sampled parametric uncertainty in a single model. However, the maximum-minimum ranges are much larger in the PPE than in the MMM at many locations. Therefore, the emulator values from PPE are more concentrated near the mean but have longer tails compared to values from MMM. This is to be expected from such a relatively small sample of models in the MMM.

Model-data comparisons are qualitatively similar to the case with MMM. The PPE simulations underestimate the observed $CCN_{0.2}$ concentrations at many stations and months. Exceptions are Puy de Dôme and Hyytiälä where PPE simulations reproduce the observations well for most of the months and Finokalia where, just like MMM, the PPE overestimates the observations. At Melpitz and Vavihill simulations capture the observed values in summer but underestimate them in winter and early spring. The PPE simulations fail to capture the observed peaks in winter and early spring at Mace Head and Cabauw as well. This is unlike the case with MMM which does not show a distinct winter time underestimate (Figure 3). The qualitative agreement between PPE and MMM indicates that the perturbed parameters are those having a significant control on aerosol processes and emissions and can be used for CCN uncertainty attribution in section 5.

### 3.3 Particle number concentration and $PM_1$ aerosol chemical composition

The observed critical diameter for particle activation into CCN at 0.2% supersaturation at most of the locations in this study is around 100 nm or larger, reaching about 200 nm in spring and summer at Finokalia (Schmale et al., 2018). Therefore in Figure 6, the MMM of the simulated $N_{50}$ and $N_{120}$ are depicted together with the 25%/75% quartiles of all models that provided station data and are compared with observations. $N_{120}$ is expected to represent a significant portion of the activated particles at 0.2% or higher supersaturation. The MMM underestimates $N_{50}$ and on average NMB is -51% and NME is 55% for all stations. $N_{80}$ is not shown in this figure but follows a similar behavior as $N_{50}$ and $N_{120}$. It is not surprising that in almost all cases both the $N_{50}$ and the $N_{120}$ concentrations are underestimated (the average NMB for MMM for all stations is -50% and the NME is 54%) by a factor that is only slightly lower than the underestimation of the $CCN_{0.2}$ concentration (-50% NMB and 60% NME). It may therefore be concluded that the quantitative differences of the models in the prediction of CCN originate from the underestimation of the number concentration of aerosol particles in the relevant size ranges. Note however that the aerosol number concentration cannot be used as a proxy for CCN levels since activation of aerosols to CCN depends not only on the size distribution but also on the chemical composition of the aerosols as well as on the supersaturation that develops in clouds (e.g., Seinfeld and Pandis, 2006; Kalkavouras et al., 2019).

Figure S1 is similar to Figures 2 and 4 but shows particulate SO4, OA mass in $PM_1$ particles at the nine stations as well as model results for DU and SS. Strong seasonal variations of the SO4 mass of about one order of magnitude are observed and simulated at the alpine site, Jungfraujoch, and at the coastal background stations, Mace Head and Finokalia, although winter minima are overestimated by the models at these coastal sites. Smaller and no clear seasonal variation of SO4 is observed at the boreal forest environment of Hyytiälä, the rural background station Cabauw and at the highly polluted Melpitz station during the year. At these three stations, the MMM underestimates the observed annual mean concentration of SO4. Strong seasonal variations of the OA mass are observed and simulated at Mace Head, Finokalia, Jungfraujoch and Hyytiälä, while no distinct seasonal cycle in organic mass is seen at Cabauw and Melpitz. The MMM is underestimating OA concentrations at all sites. The IoA between the MMM and the observations is between 0.28 and 0.62 for all stations. A detailed analysis of each model separately (Supplementary Table S6) shows that the OA mass concentration is underestimated (mean NMB is -37%) by nine of the models and overestimated by six of them (range of NMB -97% to 216%). Because different models are appearing as outliers at each station, it is difficult to conclude whether the parameterizations in one model are better than another. This, however, is consistent with the findings of a recent OA intercomparison study that considered 31 models (Tsigaridis et al., 2014) and several modeling studies that suggest a missing source of OA needed to reconcile observations with model results (Spracklen et al., 2011a; Heald et al., 2011). It appears therefore that in addition to the aerosol number concentration earlier discussed, a possible source of error in the simulation of aerosol and CCN number concentrations in the present study originates from the underestimation of the submicron OA mass at the stations where significant contribution of the submicron OA mass to the $CCN_{0.2}$ levels has been observed (Schmale et al., 2018). The importance contribution of OA in the uncertainty of CCN is also supported by the PPE simulations further discussed in section 5.

### 3.4 CCN persistence

The above analysis of CCN and aerosol number concentrations shows that on average the models are able to simulate the seasonal variations in CCN concentrations; while the model-to-observation differences in the CCN concentrations can be attributed mainly to a systematic underestimation of the number of aerosol particles that are large enough to act as CCN. The ability of models to simulate short-term variations (order of days) of the CCN number concentration is examined based on the calculated persistence of $CCN_{0.2}$ number concentrations during summer and winter (see section 2.4) for all stations and for each model. The average persistence times for all models are compared in Figure 7 with those derived from the observations (Schmale et al., 2018). Depending on the season and the station, the persistence time varies from a few hours (e.g., summer in Mace Head) to several days (e.g., winter in Melpitz).

Depending on the station, the persistence time is longer during winter (5 stations) than during summer (4 stations). The average persistence of the $CCN_{0.2}$ number concentrations simulated by the individual models is consistent with the observed change between winter and summer at 6 among the 9 stations. The models show much smaller ratio than the observations at most of the stations except at Mace Head, Noto Peninsula and Vavihill where the ratio is opposite. For the high-altitude stations, Puy de Dôme and Jungfraujoch, the models calculate longer persistence times during summer than during winter, in agreement with the observations. For these two high-altitude stations, a significant increase in the number concentration of $CCN_{0.2}$ is observed during summer, because the stations are subjected to the boundary layer air mass influence during that season, while during winter they are largely in the free troposphere. Therefore, despite the fact that the number concentration of $CCN_{0.2}$ is overall underestimated, the models are able to reproduce the dynamical behavior of these continental background stations, most probably because they are able to simulate the local meteorological changes that drive CCN persistence (supplementary Figure S4 and further discussion in supplementary section S3.1).

Analyzing the reasons that affect the persistence and then attributing the differences between the observed and the model-derived values to the underlying physical and/or chemical process parameterizations in each model is a demanding

task which is also likely to be model and case dependent. In addition to atmospheric transport patterns, dry and wet deposition processes are presumably affecting the persistence time. Because the present exercise was not focusing on deposition of aerosols, it does not have the necessary elements to elaborate on differences in the results associated with differences in the deposition parameterizations.  However, earlier global model comparisons provide insight to such differences.  Tsigaridis et al. (2014) comparison of thirty-one global models among which those participating in the present study has shown that the representation of aerosol microphysics in the models was important for dry deposition. In particular, they have shown that the use of M7 aerosol microphysics module was associated with low dry deposition fluxes of organic aerosol, which is mainly fine aerosol in the models, and the dry deposition rate coefficient ranged from 0.005 to 0.13 day$^{-1}$, i.e.  with a max/min ratio of 26. They also found that the effective wet deposition rate coefficient in the 31 participating models ranged from 0.09 to 0.24 day$^{-1}$, i.e. with a max/min ratio of 2.6 that is 10 times lower than for dry deposition, and found virtually no change between AEROCOM phase I and AEROCOM phase II models. Kim et al. (2014) compared the deposition of dust, which is mainly coarse aerosol, calculated by a smaller subset (5) of AEROCOM models. They pointed out that the size distribution of dust differs among these models and found a 30% difference in the effective dry deposition rate coefficient and about the same in the total deposition rate varying from 0.28 to 0.37 day$^{-1}$. Kim et al. (2014) analysis also revealed differences in the annual precipitation rate and in its seasonal distribution in the models and a factor of 2 differences in the fraction of wet to the total deposition of dust among the models (ranging between 0.36 and 0.63). In addition the PPE results (see section 5) clearly show that dry deposition is one of the major factors of uncertainty in the calculations of CCN in 0.2% supersaturation. Kristiansen et al. (2016) investigated the causes of differences in aerosol lifetimes within 19 global models by making use of an observational constraint from radionuclide measurements and found largely underestimated accumulation-mode aerosol lifetimes due to too fast removal in most models. In particular, they found that the way aerosols are transported and scavenged in convective updrafts makes a large difference in aerosol vertical distribution and lifetimes, as revealed in their simulations from the same model (CAM5) but with different convective transport and wet removal treatments (Wang et al., 2013) .

Furthermore, the size of the emitted OA and BC particles has been shown to be an important parameter to which the persistence time and in particular the summer-to-winter ratio of CCN is sensitive (see sensitivity runs performed with one (TM4-ECPL) among the participating models in the supplementary material; supplementary section S3.1 and Figure S5). Section 5 further attributes CCN$_{0.2}$ uncertainty to various parameters.

### 3.5 Cloud droplet number concentration from CCN spectra

Inside a cloudy updraft, $s_{max}$ is reached when supersaturation generation from expansion cooling becomes equal to its depletion by the condensation of water vapor onto the growing droplets (Nenes and Seinfeld, 2003). Increasing updraft velocity enhances the cooling rate of the cloudy air parcels, which in turn allows higher supersaturation and eventually increases $s_{max}$ and CDNC (N$_d$ in the following text and figures). Increases in CCN concentrations tend to increase N$_d$ and associated water vapor depletion in the early stages of cloud formation; this in turn hinders the development of supersaturation and implies an eventual decrease in $s_{max}$. This water vapor "competition effect" is especially strong when clouds form in the presence of large, hygroscopic particles such as sea-salt aerosol or large amounts of accumulation aerosol (Morales Betancourt and Nenes, 2014; Ghan et al., 1998). Competition effects in turn explain why droplet number responses exhibit a sublinear response to modulations in CCN; only when CCN concentrations are very low (or updraft velocities very high), $s_{max}$ becomes high enough so that the sensitivity of N$_d$ to CCN approaches unity.

Based on the behavior described above, one can understand the N$_d$ predicted from simulated and observed CCN spectra. This is straightforward for Jungfraujoch and Mace Head stations. For Cabauw and Vavihill the observed-to-simulated ratio turns from a substantial overestimation in CCN$_{0.2}$ to an underestimation in $N_d$, and the opposite is found for Finokalia. This can be

explained as follows. At both Cabauw and Finokalia, $s_{max}$ derived from observations is very low (approaching in the summer 0.07% at Finokalia and 0.04% at Cabauw; Fig. 8). The models overestimate these low values of $s_{max}$ and such values are indicative of the presence of large particles (>250nm) with sufficient hygroscopicity at these stations that are not captured by the models. Indeed, at Cabauw the available observations of CCN at 0.1% supersaturation show a larger underestimate by the models than for $CCN_{1.0}$ and $CCN_{0.2}$ (supplementary figure S16), also pointing to a model underestimate of the largest particles (>250nm) which induce the very low $s_{max}$. The overestimate in $s_{max}$ leads to an underestimate in $N_d$ by the models for all seasons except winter at Cabauw when the models at high updraft velocity capture the observationally-derived $N_d$ levels. Furthermore at Finokalia, $CCN_{1.0}$ is underestimated year-around, indicating that, in addition to the largest particles, the very small particles (smaller than 50 nm) that activate at 1.0% supersaturation and/or their hygroscopicity are also underestimated by the models there. On the other hand, particles larger than 120 nm that activate at 0.2% supersaturation are overestimated especially in winter and slightly underestimated in summer. Therefore the global models have significant difficulties in capturing the aerosol size distribution and hygroscopicity at Finokalia – which in turn translate to counterintuitive discrepancies in $N_d$.

At Vavihill a somehow different behavior is found; the underestimate of CCN at supersaturations of 0.2% and 0.7% changes to an overestimate at supersaturation 0.1% mainly in summer (supplementary Figure S16), indicating an underestimate of fine particles and/or their hygroscopicity and an overestimate of the largest particles and/or their hygroscopicity in particular during summer. This agreement of model results with observations during winter and the overestimate of CCN at 0.1% supersaturation during summer can explain the similar behavior of modelled $N_d$.

The difference between model and observationally-derived $\partial Nd/\partial w$, clearly supports the above statements. Since observations predict a suppressed $s_{max}$ compared to model distributions (Fig 8), water vapor competition effects in the observations are much more severe than in the model, indicating that observations are much more (positively) sensitive to updraft velocity. The opposite trends are seen for activation fraction ($\partial Nd/\partial Na$), given that reductions in aerosol reduce competition effects. The reduced water vapor competition effects at higher updraft velocities and the trend in CCN error also generally explain why the sensitivities are smaller for the highest updraft velocity.

As expected, both $s_{max}$ values and $N_d$ for all observations and simulations are higher for $w = 0.6\ ms^{-1}$ than for $w = 0.3\ ms^{-1}$. The response of $s_{max}$ and $N_d$ to increasing $w$ also depends on the activated fraction (Fig. 8 third row). The calculated $N_d$ increases progressively from the low values seen for the clean marine conditions at Mace Head and the high alpine atmospheric conditions of Jungfraujoch to the rural background conditions at Cabauw and Vavihill; while at Finokalia the observationally derived $N_d$ are the largest among the five stations (Fig. 9a). At Jungfraujoch, Finokalia and Mace Head, the seasonal variability of $N_d$ is captured, despite the fact that the multi-model median tends to underestimate the observationally derived $N_d$. However, the individual models show both over- and under- predictions of the observations (supplementary Fig. S3). Owing to the water vapour competition effect, $s_{max}$ decreases for increasing $N_d$, meaning that clouds at a given location do not have a "characteristic $s_{max}$", but rather depends on the given set of aerosol and dynamical conditions that develop during the cloud formation.

For all stations except Finokalia, the agreement between the model and observationally derived $N_d$ (Fig. 8) tends to be better than for CCN (Fig. 2, 4) and aerosol number concentrations (Fig. 6) (as expressed by the MMM NMB and NME for all stations provided in Table S6). Indeed, for all stations except Finokalia, NMB and NME of the MMM for $N_d$ vary from -7% to -17% and 41% to 42% respectively, with the lowest values calculated for the low updraft velocity. For $CCN_{0.2}$ NMB is -59% and NME 63%, averaged over the same stations. This trend is a result of the competition effect of CCN on $s_{max}$; if observed CCN concentrations are higher than predicted, then the "observed" $s_{max}$ tends to be less than the "predicted" $s_{max}$ - which means the discrepancy in "observed" and "predicted" $N_d$ is reduced compared to the CCN errors. The error reduction is substantial, especially under lower updraft velocity conditions. As qualitative example we here present the ratio of the observed to the simulated average values of $CCN_{0.2}$ number concentrations that is 4.0 at Jungfraujoch, 2.2 at Cabauw, 2.1 at

Mace Head, 1.5 at Vavihill, and 0.8 at Finokalia (Fig. 3). In the case of $N_d$ the corresponding ratios for $w=0.6$ m.s$^{-1}$ are ~1.8 at Jungfraujoch, ~0.9 at Cabauw, ~1.5 at Mace Head, ~0.9 at Vavihill and ~1.8 at Finokalia (Fig. 9). All these ratios are inversely correlated with the observed to the simulated average values of $s_{max}$ (Fig. 9), a clear indication of competition effects on $N_d$ and prediction error mitigation.

In agreement with our finding, Sotiropoulou et al. (2006) using a similar approach applied to observations from the ICARTT field campaign estimated that a 20–50% error in CCN closure results in a 10–25% error in $N_d$, while global simulations suggest global average CCN prediction error between 10 and 20% and a smaller corresponding $N_d$ error between 7 and 14% (Sotiropoulou et al., 2007). Such reduction in error can be explained by a self-regulation by $N_d$ since $S_{max}$ decreases with increasing aerosol number concentration as discussed by many studies published to date (e.g., Twomey et al.,

1959; Charlson et al., 2001; Nenes and Seinfeld, 2003; Feingold and Siebert, 2009), giving rise to regions where $N_d$ is relatively insensitive to changes in CCN or updraft velocity (e.g., Rissman et al., 2004, and Reutter et al., 2017). At very high CCN levels, and in the presence of sufficient large hygroscopic CCN, $N_d$ may actually decrease with increases in aerosol amount (Ghan et al., 1998; Feingold, 2001; McFiggans et al., 2006; Reuter et al., 2017); parameterizations that do not fully capture these important aspects of the aerosol-droplet relationship may also give rise to biases in aerosol indirect

forcing assessments (e.g., Morales-Betancourt et al., 2014a).

These results clearly indicate that the number of CCN at a prescribed supersaturation cannot be used as an indicator for the number of activated droplets. The maximum supersaturation that develops inside the cloud (hence droplet number) responds to changes in aerosol and vertical velocity levels and thus is dynamically determined and can vary considerably for a given site. This is even further complicated by the potential for model biases to change even sign across at cloud-relevant

supersaturations. CCN-derived comparisons cannot even be used qualitatively, as the supersaturation levels can be so different from a prescribed value that even the error trend in $N_d$ may not be reflected. For example, according to observationally derived data, CCN$_{0.2}$ at Cabauw is significantly higher than at Finokalia, although at Finokalia $N_d$ is larger for the observations but not for the model results. Our analysis however clearly shows that the models examined here do not exhibit the same level of $N_d$ prediction error as CCN error– a robust trend which is a result of the physics of cloud droplet

formation. Because of the discrepancy in sensitivities $\partial Nd/\partial Na$ and $\partial Nd/\partial w$, models may be predisposed to be too "aerosol-sensitive" or "aerosol-insensitive" in aerosol-cloud-climate interaction studies, even if they may capture average droplet numbers well. This is a subtle, but profound finding that only the sensitivities can clearly reveal and may explain inter-model biases on the aerosol indirect effect. Few published efforts (apart from Morales et al., 2014a and Sullivan et al., 2016) can demonstrate this, but none over a range of models and using a considerable aerosol dataset for evaluation as here performed.

## 4 Global distributions of surface CCN$_{0.2}$ and particle number concentrations

The global near-surface annual mean MMM distributions of the $N_3$, $N_{50}$ and CCN$_{0.2}$ number concentrations for the year 2011 (Fig. 10) show similar patterns, i.e. larger concentrations over the continents due to the primary anthropogenic emissions over industrialized areas in USA, Europe, and Asia, and dust and biomass burning emissions in the tropics.

Multi-model median near surface $N_3$ number concentrations over continental regions vary between 1,000-10,600 cm$^{-3}$, while over the marine boundary layer (MBL) they vary between 100-2,000 cm$^{-3}$, rarely exceeding 300 cm$^{-3}$ (Fig. 10a). The MMM $N_3$ surface distribution is similar to the results by Spracklen et al. (2010) and Gordon et al. (2017), who computed maximum $N_3$ concentrations of ~10,000 cm$^{-3}$. The concentration of $N_3$ is directly related to new particle formation and growth as well as to primary emitted particles. Since models use different nucleation mechanisms and emission inventories it

is expected that the diversity of the model results is higher for $N_3$ than for particle number concentration with larger (low-end) cut-off diameter. The largest diversities in the model results (Fig. 10b) are found in the polar regions, where

concentrations are relatively low, and in the continental boundary layer with high values (about 2) observed in the tropics and particularly in South America and over the boreal regions in Asia. Diversities of up to 1.5 are computed for the Mediterranean, Arabian Peninsula, Central Africa, Indonesia and South East Asia, indicating differences between models in the representation of primary and secondary aerosol sources in these regions. Over the oceans the diversity is lower (<1) except in the high latitudes of the Northern Hemisphere where it exceeds 1.5. Even lower model diversity (around 0.8) is found in highly polluted areas over North America and Europe indicating consistency between models, in the representation of aerosols in these regions. In addition to new particle formation, our results point mainly to biomass burning emissions as major source of uncertainty in the model calculations resulting in high model divergence in areas like southern Europe, tropical Africa and America, Southern Asia and Indonesia. Assumption of emission injection height is also a source of discrepancy between models, leading to differences in the calculated lifetimes (up to 30%) and in the tropospheric columns (up to 25%) of pollutants (Daskalakis et al., 2015), while differences of an order of magnitude in their concentrations are computed for the middle troposphere (Jian and Fu, 2014). Thus, differences in the emission injection heights in the participating models, as outlined in section 2.1 and Table S4, contribute to the model results divergence. The highest maximum $N_3$ concentrations in a $5^o$x$5^o$ grid box (supplementary Figure S6) were computed by the GISS-E2.1-MATRIX model (~176,000 cm$^{-3}$) and the TM4-ECPL model (~102,000 cm$^{-3}$) while the lowest were from the ECHAM6_HAM2-AP model (~6,400 cm$^{-3}$). A sensitivity simulation was performed by a single model (TM4-ECPL; discussed in sect. 3.3 and supplementary section S3.1 and Figure S5) assuming the same primary emissions of carbonaceous aerosol in terms of mass to be emitted at larger particle sizes. This additional simulation shows the importance of the assumptions on size distribution of the emissions in the models since the results of this simulation are very close to the average of the other models. In agreement with these findings, Spracklen et al. (2010) concluded that the sensitivity of $N_3$ to the size of emitted particles originating from anthropogenic activities is significantly higher in regions close to anthropogenic sources and significantly lower at the remote boundary layer sites.

The annual global mean distribution of near-surface $N_{50}$ particle number concentrations (Fig. 10c) is similar to that of the $N_3$ particles, but the number concentrations are lower for these larger particle sizes that are more relevant for CCN. The spatial distributions of $N_{50}$ are similar, but their concentrations are reduced by about a factor of 2.5 compared to $N_3$. The highest values of $N_{50}$ are found over or close to industrialized regions due to anthropogenic emissions, and over Central Africa and South America due to strong biomass burning emissions. Over marine regions, $N_{50}$ is higher in the Northern Hemisphere than in the Southern Hemisphere due to the outflow from continental anthropogenic sources. Despite the similarities of the global MMM distributions, the models' diversity and spatial pattern of $N_{50}$ (Fig. 10d) differ significantly from that of $N_3$. Excluding polar regions as for $N_3$, the highest model diversities for $N_{50}$ (~2) are observed in regions with strong biomass burning emissions (Southern America, Central Africa and Indonesia) and high diversities are also found over the tropical Pacific which might be associated with marine emissions representation in the models. In all other regions the diversity of $N_{50}$ simulations does not exceed 1, even over the remaining tropical and southern oceans, where sea salt is important.

The near surface MMM concentrations of the $CCN_{0.2}$ do not exceed 3,500 cm$^{-3}$ over polluted areas in Europe, Asia and the United States, as shown in Fig. 10e. This value is in the range of the 3,162-10,000 cm$^{-3}$ $CCN_{0.2}$ concentrations simulated by Spracklen et al. (2011) over China and attributed to carbonaceous aerosols acting as CCN. In the present study, only one model (EMAC) computes $CCN_{0.2}$ levels that exceed 10,000 cm$^{-3}$ over the Taklimakan desert in Asia, while the remaining fourteen models show maximum surface $CCN_{0.2}$ concentrations < 5,000 cm$^{-3}$ (see supplementary Figure S9). The surface distribution and magnitude of $CCN_{0.2}$ is similar to $N_{120}$ (supplementary Figure S8) with the maximum $CCN_{0.2}$ concentrations only slightly lower than the $N_{120}$ values for most models, indicating that most of the $N_{120}$ particles activate, implying a global-mean kappa of ~0.2 for 120 nm particles. However, analysis of the individual model results over the polluted areas shows that the number concentration of $N_{120}$ can, in most cases, be either 50% lower or higher than that of $CCN_{0.2}$. The

modeled $CCN_{0.2}$ diversity is lower than the diversity for $N_{50}$ with values < 0.5 for mid-latitude continental regions and around 1 over the tropical oceans, where the $CCN_{0.2}$ number concentration is usually lower than 60 $cm^{-3}$, but also over the tropical S Africa and Central Africa where $CCN_{0.2}$ number concentration is a few hundreds of $cm^{-3}$. $CCN_{0.2}$ model diversity is also lower than that of $N_3$ simulations. The maximum reduction of the model diversity in $CCN_{0.2}$ simulations compared to that in $N_3$ simulations is found to exceed a factor of 9 and maximizes over the high latitudes of the North Hemisphere and the South Arabian Peninsula where new particle formation is high. Overall, a global mean reduction of a factor of about 2 is found as shown in Fig. S18 that provides the ratio of $N_3$ model diversity (Fig 10b) over $CCN_{0.2}$ model diversity (Fig. 10f).

Some of the differences in global near surface distributions of CCN (Fig. S9) can be associated with the corresponding differences in the computed SO4 and OA surface distributions (Fig. S10 and Fig. S11, respectively). For instance, in China and S. America, models that are biased low in SO4 and high in OA are also biased low in CCN. Significant differences are also found for black carbon, sea-salt and dust $PM_1$ components (Fig. S12-S14). In particular, for all models near-surface BC distributions maximize over China, while individual models differ by a factor of 3 to 4. Simulated SS distributions maximize over the southern oceans where the models show the largest differences up to 2 orders of magnitude reflecting large differences in the parameterized emissions of SS (see also supplementary Table S3). Finally, DU distributions show the largest spread among models with near surface values that differ up to a factor of 40. The global surface distributions of the MMM of the chemical compound (SO4, BC, OA, SS and DU) concentrations that contribute to $PM_1$ are shown in the supplementary Figure S15 (left column) together with the corresponding model diversities (right column). For all simulated $PM_1$ components diversities maximize south of 60°S and north of 60°N, similarly to $N_3$, which reflects the challenges of the models in simulating atmospheric transport, deposition and chemistry close to the poles.

## 5. Causes of uncertainty in CCN

In this section we use the HadGEM-UKCA perturbed parameter ensemble (PPE) to identify some potential causes of model diversity and bias compared to the observations. We performed a variance-based sensitivity analysis at each measurement site using the 260,000 HadGEM-UKCA model variants sampled from the emulator following the methodology described in previous studies (Lee et al., 2013; Johnson et al., 2018).

Figure 11 shows the fraction of variance in $CCN_{0.2}$ that can be attributed to each of the perturbed parameters. Here we draw attention to the main parameter effects and refer to Yoshioka et al. (2019) for a full description of all parameters. The list of these parameters is provided in the caption of Fig. 11. In the summer, the largest contributions to uncertainty in $CCN_{0.2}$ at most sites are the biogenic volatile organic compounds (BVOC) emission flux and the assumed hygroscopicity of the organic matter in the particles ($\kappa_{OA}$). The BVOC emissions in this model are assumed to be $\alpha$-pinene and to have an uncertainty range of 12-225 Tg SOA production per year. The $\kappa_{OA}$ is assumed to have a range of 0.1-0.6 and to be fixed during the simulation time (i.e., the hygroscopicity does not change due to within-particle oxidation). Together, these two mostly biogenic-related parameters account for up to 90% of the CCN variance in summer, ranging from about 0% at Mace Head, 20% at Cabauw, 40% at Finokalia, 70% at Melpitz, to 90% at Hyytiälä. These results show that at Hyytiälä the organic fraction of CCN-active aerosol is highest, while at other locations, like Mace Head, the inorganic fraction dominates the total hygroscopicity. Except at the Mace Head coastal site, the other important parameters in summer are dry deposition of aerosol, anthropogenic $SO_2$ emissions (at Finokalia, Puy de Dôme and Jungfraujoch), the fossil fuel emission flux (at Noto Peninsula, Cabauw and Melpitz), and the assumed width of the accumulation mode (at Jungfraujoch and Puy de Dôme).

In winter, aerosol dry deposition is an important cause of uncertainty in $CCN_{0.2}$ at all sites except Jungfraujoch and Puy de Dôme. At most sites (except Mace Head and Noto Peninsula) the emissions fluxes (and the assumed particle sizes) of

carbonaceous aerosol from fossil fuel and residential combustion sources account for 10-20% of the uncertainty. Ageing of aerosol through uptake of sulphuric acid and SOA is also important at these sites. Finally, the production of sulphate through in-cloud oxidation by ozone (perturbed parameter marked as 'Cloud pH') accounts for 30-40% of the uncertainty at Finokalia, Puy de Dôme and Jungfraujoch.

In summary, the PPE results suggest that production of SOA from biogenic emissions combined with the hygroscopic properties of the OA should be investigated as a source of differences in predicted CCN between models in summer. In winter, dry deposition, ageing and in-cloud sulphate production are the dominant sources of CCN uncertainty. Given that the importance of CCN prediction uncertainty may not always translate to CDNC uncertainty – especially if cloud formation occurs in a velocity-limited regime - any future analysis should place CCN uncertainty within the context of CDNC uncertainty.

## 6 Summary and conclusions

Within the BACCHUS/AEROCOM multi-model CCN intercomparison initiative, a total of 16 global aerosol-climate and chemistry-transport models have compared to each other and to observations. Among them 14 provided results for particle and CCN number concentrations and $PM_1$ component mass concentrations, which have been compared to surface observations at eight sites in Europe and one in Japan to evaluate the skill of the simulations.

In this inter-model comparison, models used different meteorology and emissions (e.g., CMIP5/6) datasets/parameterizations. Most models (including the multi-model median) tend to underestimate the observed aerosol number concentrations $N_{50}$, $N_{80}$ and $N_{120}$, as well as the CCN concentrations, suggesting incomplete understanding of the underlying processes. In particular, emissions and the size distribution of emitted particles, injection heights of biomass burning emissions, atmospheric ageing and particularly aqueous phase chemistry, the hygroscopicity of organic aerosol, and dry and wet deposition have been pointed out as main sources of uncertainties in model simulations. Models are, however, reproducing between 45% and 86% of the seasonal variability of $N_{50}$, $N_{80}$, $N_{120}$, and $CCN_{0.2}$ number concentrations, and SO4 and OA $PM_1$ component mass concentrations, with the exception of Hyytiälä where only 36% of the SO4 variability is captured by the MMM, as indicated by the correlation coefficient of the MMM with the observations (Table S6). While models are improved since the 2014 AEROCOM organic aerosol intercomparison (Tsigaridis et al., 2014), most continue to underestimate the organic submicron aerosol mass concentrations. Thus the MMM underestimates observed OA $PM_1$ mass concentrations by 36% (for Hyytiälä) to 77% (for Jungfraujoch).

The simulated $N_3$ number concentrations, which are generally higher over land, show high diversity among models over the Northern Hemisphere continents, while the simulated CCN are less diverse. Overall a global mean reduction of a factor of about 2 is found in the model diversity in $CCN_{0.2}$ simulations compared to that in $N_3$ simulations, maximizing over regions where new particle formation is important. This finding points to differences in the size distribution of the primary emissions and/or in the formation and growth of new particles as important sources of the inter-model diversity in CCN.

CCN number concentrations are generally underestimated at all supersaturations by the MMM by at least 34% (Figure 9, Table S6), with the exception of very low supersaturations, indicating that models have most difficulty in capturing the largest particles (>250 nm) that activate at very low supersaturations. There is no model that performs best at all stations. The models on average qualitatively capture the strong seasonal variabilities of CCN observed at Finokalia, Noto Peninsula, Puy de Dôme and Jungfraujoch, and the very weak seasonality observed at the other stations. Production of SOA from biogenic emissions combined with the hygroscopic properties of the OA in summer and dry deposition, ageing and in-cloud sulphate production in winter have been identified by PPE simulations as dominant sources of CCN uncertainty and should in future be investigated.

The short-term variability of $CCN_{0.2}$ at the measurement sites has been examined by comparing the $CCN_{0.2}$ persistence time computed from the observed data and the model results. Because persistence time is a normalized timescale, driven by the processes that "set" the CCN concentrations, it is more sensitive to air mass changes and the formation/removal rates of atmospheric particles than to the exact number concentration of CCN. With the exception of two models that estimate very large persistence times (about 16 days) during summertime at Finokalia, the modeled persistence times of near-surface $CCN_{0.2}$ are between 0.5 and 9 days depending on the model, location and season (Figure S4), range similar to that derived from observations that vary between about 0.5 and 7 days. At 6 out of 9 stations the average relative change in modeled persistence time between winter and summer is in agreement with observations. These persistence times of $CCN_{0.2}$ are sensitive to assumptions on size of the emitted particles, as shown by a sensitivity simulation with TM4-ECPL model.

A novel aspect of this study is the comparison of ensemble global aerosol climate model near-surface results with experimentally derived CDNC from surface measurements of CCN at different levels of supersaturation. Note that CDNC is not calculated by each participating model but a common methodology has been followed to derive the CDNC from the modeled and observed CCN spectra. Despite the large differences between models and observations found in the number concentration of aerosol particles and CCN, the CDNC estimates based on the CCN spectra are in significantly better agreement than the CCN for the stations examined here. In addition, the inter-model spread of CDNC is smaller than that of particle and CCN number concentrations. These trends are robust and a result of the physics of cloud droplet response to aerosol perturbations and show a self-regulation by CDNC.

As for CCN number concentrations, in several cases models underestimate CDNC when compared to the observationally derived CDNC (section 3.5). At high aerosol number concentrations, the maximum supersaturation is computed to be low, limiting the fraction of particles that can activate and form CDNC. As a result, the sensitivity of CDNC to updraft velocity prevails. On the contrary, at high updraft velocities, CDNC is controlled by the variability in the aerosol number concentration. An anticorrelation is found between the sensitivity of CDNC to the number of aerosols and that to the updraft velocity, showing that the variability of these two parameters can explain the variability in CDNC and limit CDNC formation.

Our results are in agreement with previous studies showing that CDNC are sensitive to the uncertainties in the CCN number concentrations mainly in regions where aerosol number concentrations are low and support the concept of existence of two distinct regimes ("aerosol-limited", and, "updraft-limited"). Unlike previous studies, however, we show that for a large number of models, persistent and substantial CCN prediction biases are considerably reduced when expressed as droplet number concentrations for boundary layer-type clouds. Biases in CDNC are found to be qualitatively different from the biases in $CCN_{0.2}$ and are attributed to the ability of models to capture the levels of the largest particles that activate at very low cloud-relevant supersaturations. These results point to the need for observations that cover the CCN spectra down to very low supersaturation levels, and demonstrate that model-observation comparisons of CCN at a prescribed supersaturation may be misleading in the error evaluation for CDNC, since supersaturation is dynamically determined and can vary considerably for a given site. The methodology proposed here, however, overcomes this limitation and considers the dynamic nature of supersaturation adjustment to CCN variations thus determining appropriate supersaturation levels for model-observation comparison. Such methodology can help better guide modeling efforts to focus on regions where CDNC predictions are most biased and sensitive to CCN perturbations (e.g., in the Southern Oceans).

*Acknowledgements.* This work has been supported by the European Union's Seventh Framework Programme (FP7/2007-2013) collaborative project BACCHUS (Impact of Biogenic versus Anthropogenic emissions on Clouds and Climate: towards a Holistic UnderStanding under grant agreement 603445. AN acknowledges support from the project PyroTRACH (ERC-2016-COG) funded from H2020-EU.1.1. - Excellent Science - European Research Council (ERC), project ID 726165.

Dirk Olivie has contributed with coding and information about the emission heights used in CAM5.3-Oslo. TB and TvN acknowledge funding from the European Union's Horizon 2020 research and innovation programme under grant agreement No 641816 (CRESCENDO) for their contributions to this study. The development of NorESM1 has been supported by the Norwegian Research Council through the projects EarthClim (207711/E10), EVA (grant no. 229771), the NOTUR (nn2345k) and NorStore (ns2345k) projects, and through the Nordic Centre of Excellence eSTICC (57001) and the EU H2020 project CRESCENDO (Grant no. 641816). PS also acknowledges funding from the European Research Council project RECAP under the European Union's Horizon 2020 research and innovation programme with grant agreement 724602. DWP acknowledges funding from Natural Environment Research Council project NE/L01355X/1 (CLARIFY). The ECHAM-HAMMOZ model is developed by a consortium composed of ETH Zurich, the Max Planck Institut für Meteorologie, Forschungszentrum Jülich, the University of Oxford, the Finnish Meteorological Institute and the Leibniz Institute for Tropospheric Research, and managed by the Center for Climate Systems Modeling (C2SM) at ETH Zurich. The computing time for this work was supported by a grant from the Swiss National Supercomputing Centre (CSCS) under project ID s652. HM acknowledges funding from  the Japan Society for the Promotion of Science (JSPS), the Ministry of Education, Culture, Sports, Science, and Technology and the Japan Society for the Promotion of Science (MEXT/JSPS) KAKENHI Grant Numbers JP26740014, JP17H04709, JP26241003, and JP16H01770, MEXT Green Network of Excellence (GRENE) and Arctic Challenge for Sustainability (ArCS) projects, and the Environment Research and Technology Development Fund (2-1403, 2-1703) of Environmental Restoration and Conservation Agency. HW and YY acknowledge support from the U.S. Department of Energy (DOE) Biological and Environmental Research and the ACTIVATE project (a NASA Earth Venture Suborbital-3 investigation) funded by NASA's Earth Science Division and managed through the Earth System Science Pathfinder Program Office. The Pacific Northwest National Laboratory (PNNL) is operated for DOE by Battelle Memorial Institute under contract DE-AC05-76RLO1830. The CAM5_MAM3 simulation was performed using PNNL Institutional Computing resources. JK and JP were funded by the U.S. National Science Foundation, Atmospheric Chemistry Program, under Grant No. AGS-1559607 and the U.S National Oceanic and Atmospheric Administration, an Office of Science, Office of Atmospheric Chemistry, Carbon Cycle, and Climate Program, under the cooperative agreement award #NA17OAR430001. GL and FY acknowledge support from NASA under grant NNX13AK20G and NSF under grant 1550816. SEB and KT acknowledges funding from NASA's Atmospheric Composition Modeling and Analysis Program (ACMAP), contract number NNX15AE36G. Resources supporting this work were provided by the NASA High-End Computing (HEC) Program through the NASA Center for Climate Simulation (NCCS) at Goddard Space Flight Center. MG acknowledges support from the ACTRIS2 project, EU H2020-INFRAIA-2014-2015, grant agreement no. 654109, the Swiss State Secretariat for Education, Research and Innovation (SERI) under contract number 15.0159-1, and MeteoSwiss in the framework of the Swiss contributions (GAW-CH and GAW-CH-Plus) to the Global Atmosphere Watch programme of the World Meteorological Organization (WMO). This work has been finalized when MK was visiting L. Gallardo at the Center for Climate and Resilience Research (CR2) at the Geophysics Department of the University of Chile where MK profited from fruitful discussions.

**Data availability**

The data used for this study are available on line at http://ecpl.chemistry.uoc.gr/BACCHUS_CCN (contact: Maria Kanakidou, mariak@uoc.gr).

**Authors contributions**

GF performed TM4-ECPL model simulations, developed codes and has drawn all figures for the multi-model data analysis. MK initiated and coordinated the exercise. AN developed the codes to interpret the CCN spectra from all models, the obser-

vations and performed CDNC calculations. MK and AN defined the experimental protocol. GF, MK, AN analyzed data and wrote the manuscript. All co-authors have contributed data and to the writing of the manuscript.

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

**Table 1.** Hygroscopicity parameters used by the participating models for water uptake calculations.

| MODEL | SO4 | OA | SS | DU | BC | NO3 |
|---|---|---|---|---|---|---|
| CAM5-chem-APM | 0.9 | 0.1 | 1.28 | 0 | 0 | 0.9 |
| CAM5-chem-ATRAS2 | 0.61 | 0.1 | 1.16 | 0.001 | 1.e-6 | 0.61 |
| CAM5_MAM3 | 0.507 | 0.1 | 1.16 | 0.068 | 0 | N/A |
| CAM5_MAM4 | 0.507 | 0 | 1.16 | 0.068 | | N/A |
| CAM5.3-Oslo | 0.507[(0)] | 0.14 | 1.2 | 0.069 | 5.e-7 | N/A |
| ECHAM5.5-HAM2-ELVOC_UH | 0.6 | 0.06 | 1.12 | 0 | | |
| ECHAM6-HAM2 [(1)] | 0.7 | 0. | 1.3 | 0 | 0 | N/A |
| ECHAM6-HAM2-AP[(1)] | 0.7 | 0 | 1.3 | 0 | 0 | N/A |
| EMAC [(2)] | | 0.1 | 1.12 | 0 | 0 | N/A |
| GEOS-Chem-APM | 0.9 | 0.1 | 1.28 | 0 | 0 | 0.9 |
| GEOS-Chem-TOMAS | 1.0 | 0.1[(3)] | 1.2 | 0.01 | 0 | N/A |
| GISS-E2.1-MATRIX | 0.507 | 0.141 | 1.335 | 0.14 | 5.e-7 | 0.507 |
| GISS-E2-TOMAS | 0.7 | 0.15[(4)] | 1.3 | 0 | 0 | N/A |
| TM4-ECPL | 0.6 | 0.1 | 1.0 | 0 | 0 | N/A |
| TM5 | 0.6 | 0.1 | 1.0 [(5)] | 0 | 0 | 0.6 |

[(0)] In CAM5.3-Oslo the hygroscopicity parameters κ for pure ammonium sulphate or sulphuric acid are 0.507 and 0.534, respectively. For internal mixtures, κ is a mass weighted average of the aerosol components, except for particles coated (> 2nm) with SO4, OA and/or SS, where κ is a mass weighted average of the components of the coating (Kirkevåg et al., 2018).

[(1)] ECHAM6-HAM2 and ECHAM6-HAM2-AP use the Abdul-Razzak and Ghan (AR-G) activation scheme (Abdul Razzak and Ghan, 2000). The reported values are approximated using the number of ions and osmotic coefficients used in the AR-G scheme.

[(2)] EMAC model simulates the effective hygroscopicity parameter κ of each aerosol size mode in order to describe the influence of chemical composition on the CCN activity of aerosol particles (Pringle et al., 2010). These values are the
internally mixed κ calculated across the nucleation, Aitken, accumulation and coarse modes. The effective aerosol hygroscopicity parameter κ is calculated according to the simple mixing rule proposed by (Petters et al., 2007) using the volume fraction and hygroscopicity parameter of each chemical component (23 salts from ISORROPIA-II and 4 bulk species) taken from (Petters et al., 2007) and (Sullivan et al., 2009)

[(3)] for hydrophilic OA κ=0.1, for hydrophobic OA κ=0.01

[(4)] for hydrophilic ORG. For hydrophobic, κ=0.

[(5)] for NaCl  κ=1, for $Na_2SO_4$ κ=0.95

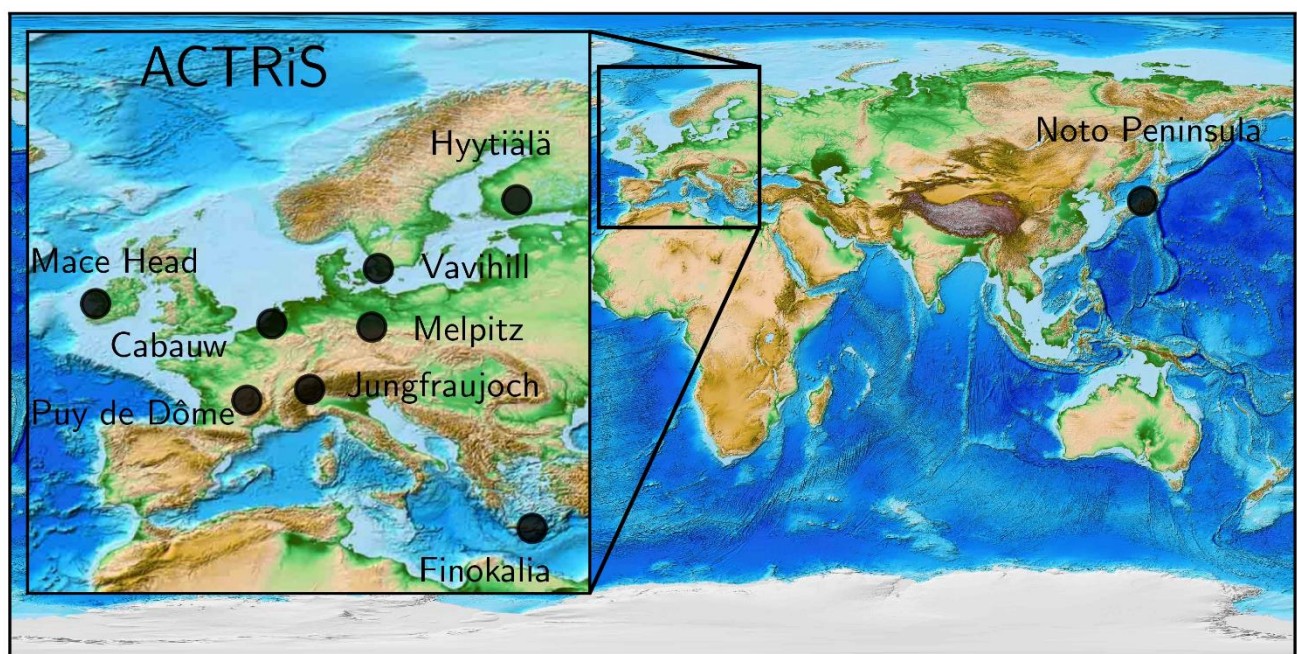

**Fig. 1.** Map showing the location of the measurements sites used in this study**.**

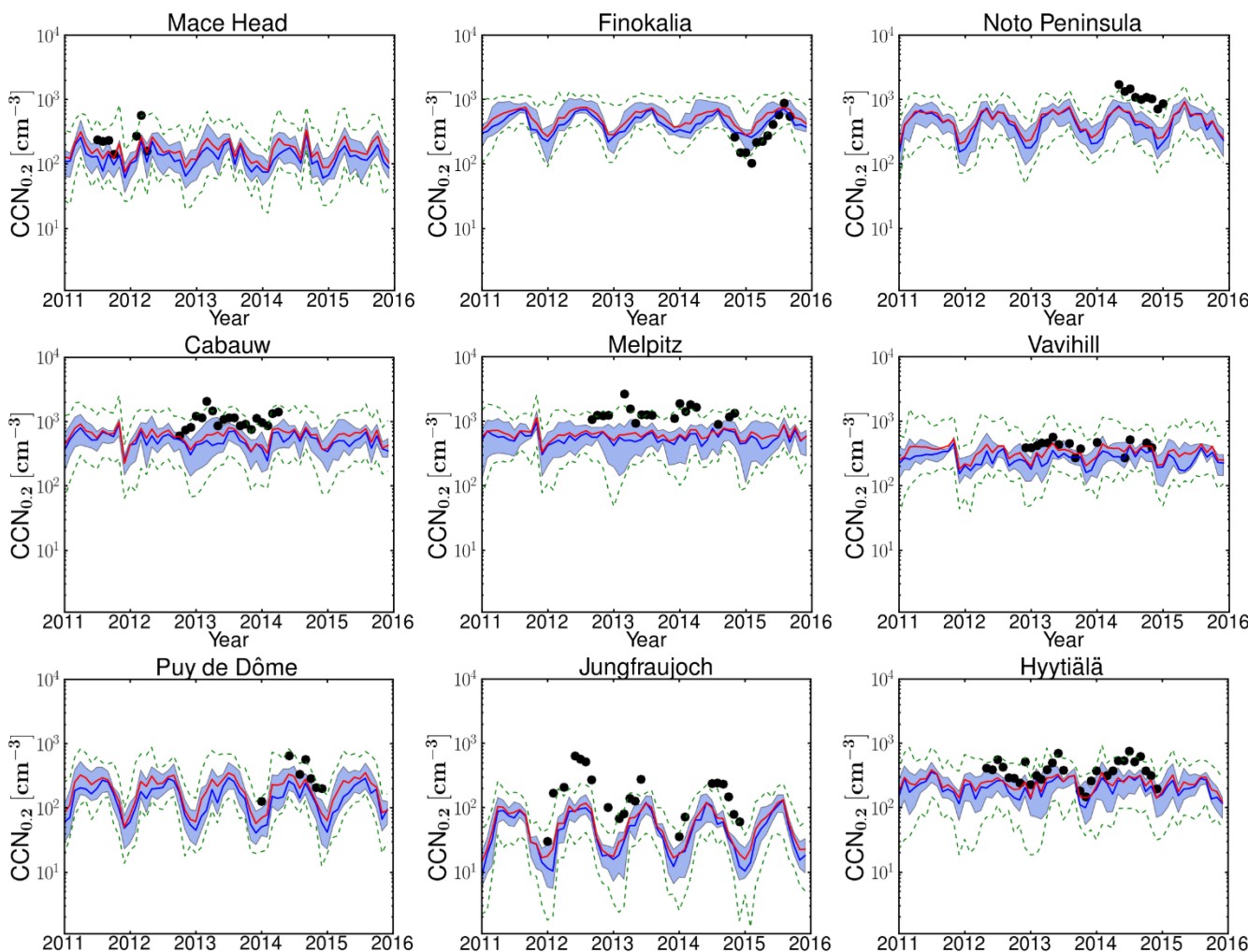

**Fig. 2.** Monthly ensembles for the years 2011-2015 of the CCN number concentration for supersaturation 0.2% ($CCN_{0.2}$). The $CCN_{0.2}$ obtained from observational data is shown with symbols. The continuous bold blue and red lines show the monthly median and mean of the all models, respectively. The shaded area shows the 25/75% of the model results, while the green dashed lines the minimum and maximum values of all models.

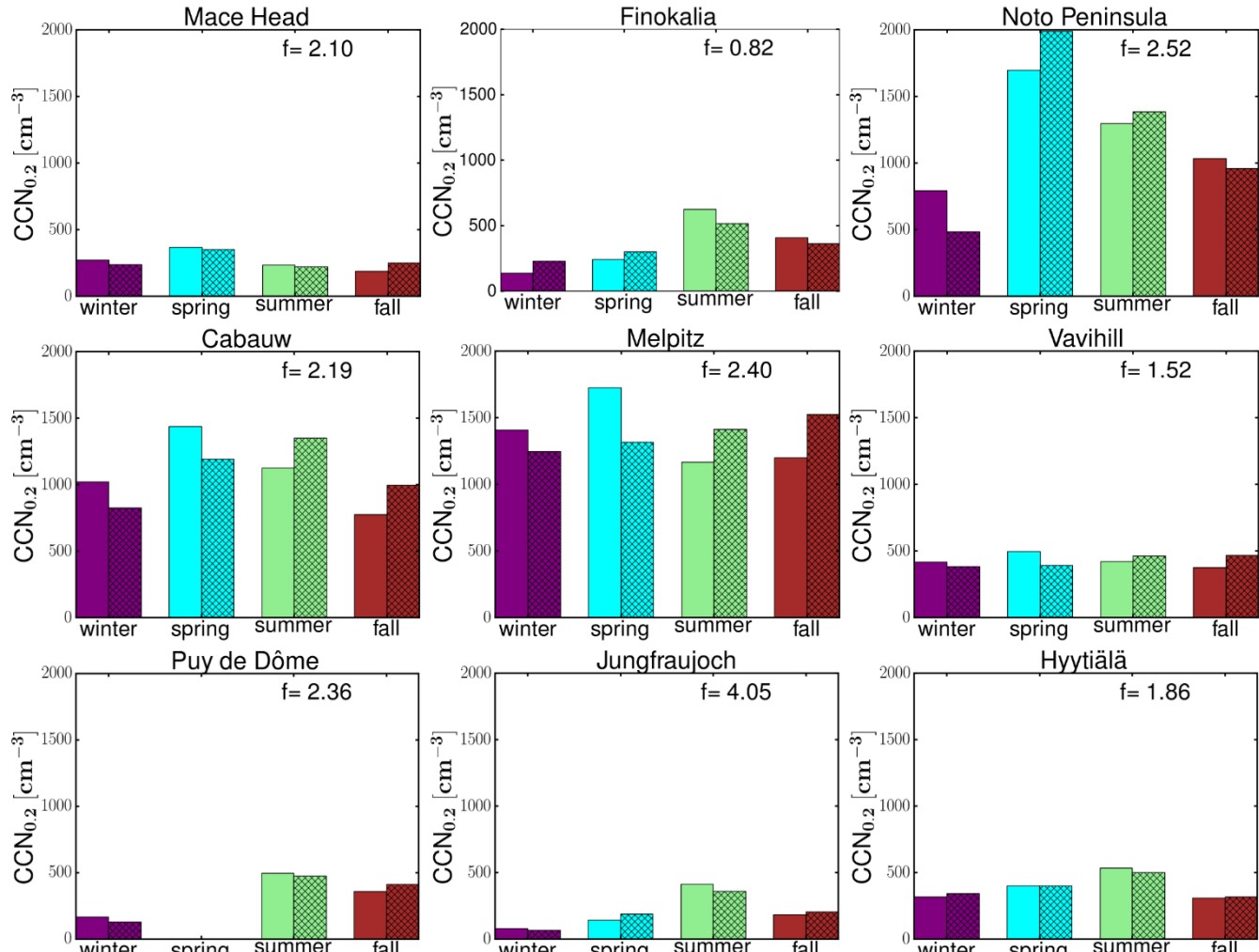

**Fig.3.** Comparison of the seasonal variations of the observed and model-median computed $CCN_{0.2}$. The solid bars show the average of the observed $CCN_{0.2}$ during each season and the shaded bars the corresponding averages of the model results. The simulated $CCN_{0.2}$ concentrations have been scaled by a factor, $f$ (denoted in each graph), so that the four seasons mean is the same as the observed one. For Puy de Dôme normalization is based on the mean of three seasons (winter, summer and fall) due to data availability.

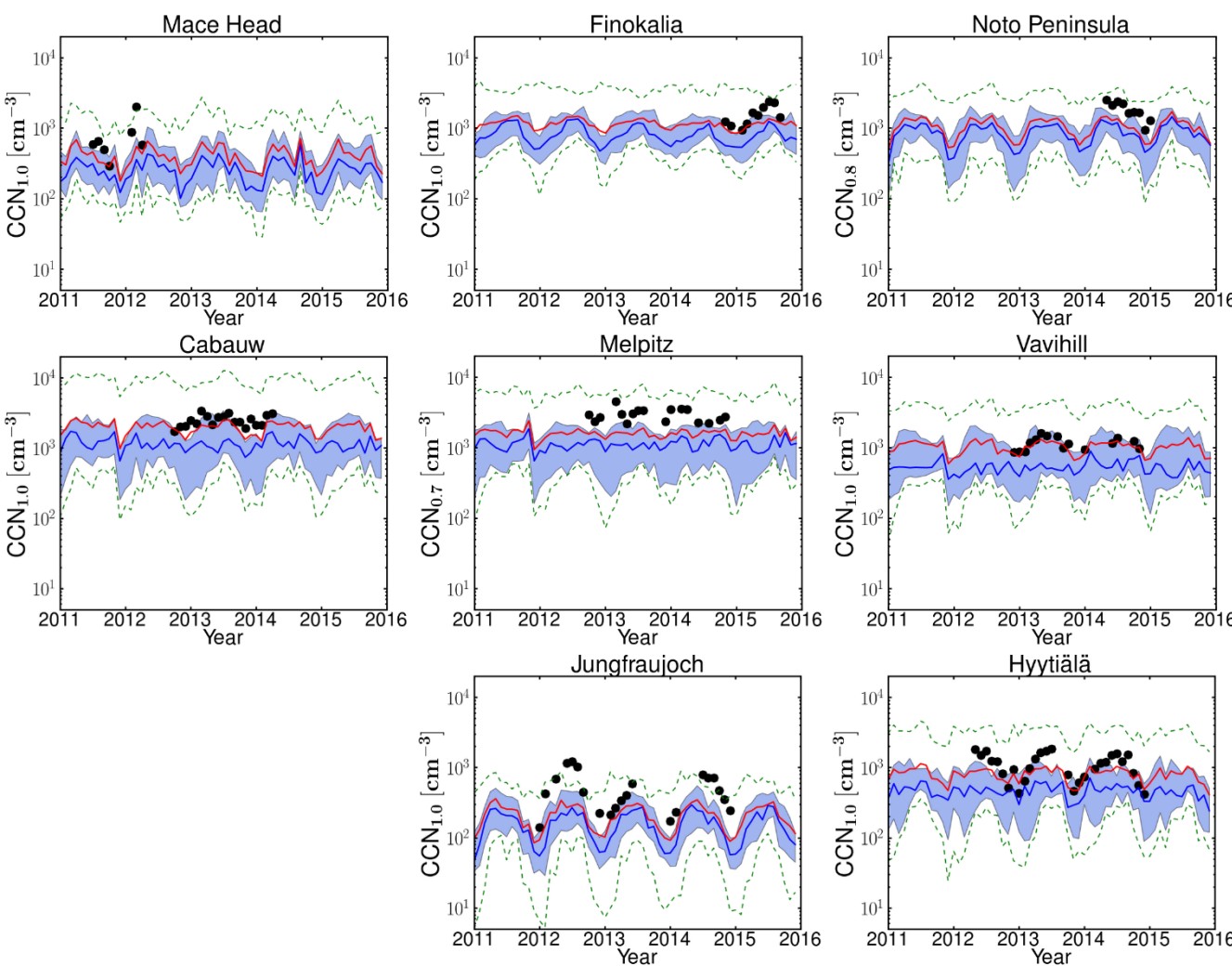

**Fig. 4.** Same as Figure 2 for the CCN at the maximum supersaturation with available measurements at each station. For Puy de Dôme only $CCN_{0.2}$ data are available and are shown in Figure 2.

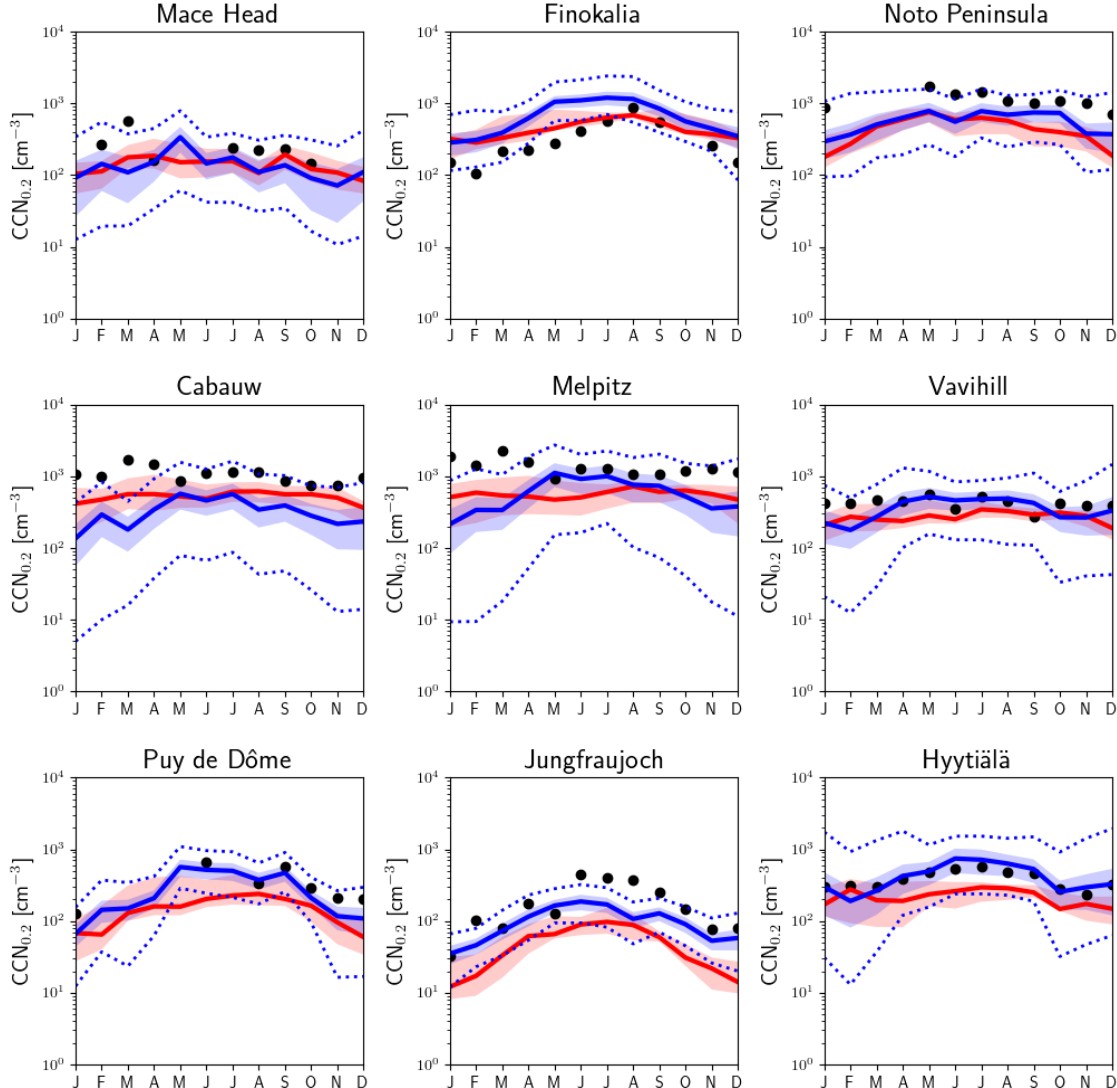

**Fig. 5.** Monthly average $CCN_{0.2}$ based on HadGEM3-UKCA perturbed parameter ensemble simulations for year 2008. The solid blue line shows the mean of 260,000 model emulators for each month and station. The shaded blue area shows the range of emulator mean plus and minus one standard deviation, while the blue dashed lines show the minimum and maximum emulator values. The red line shows the MMM results (mean of the years 2011-2015 shown in Figure 2) and the shaded red area corresponds to the 25%/75% quartiles. The $CCN_{0.2}$ obtained from observational data are shown in symbols (mean of the available data).

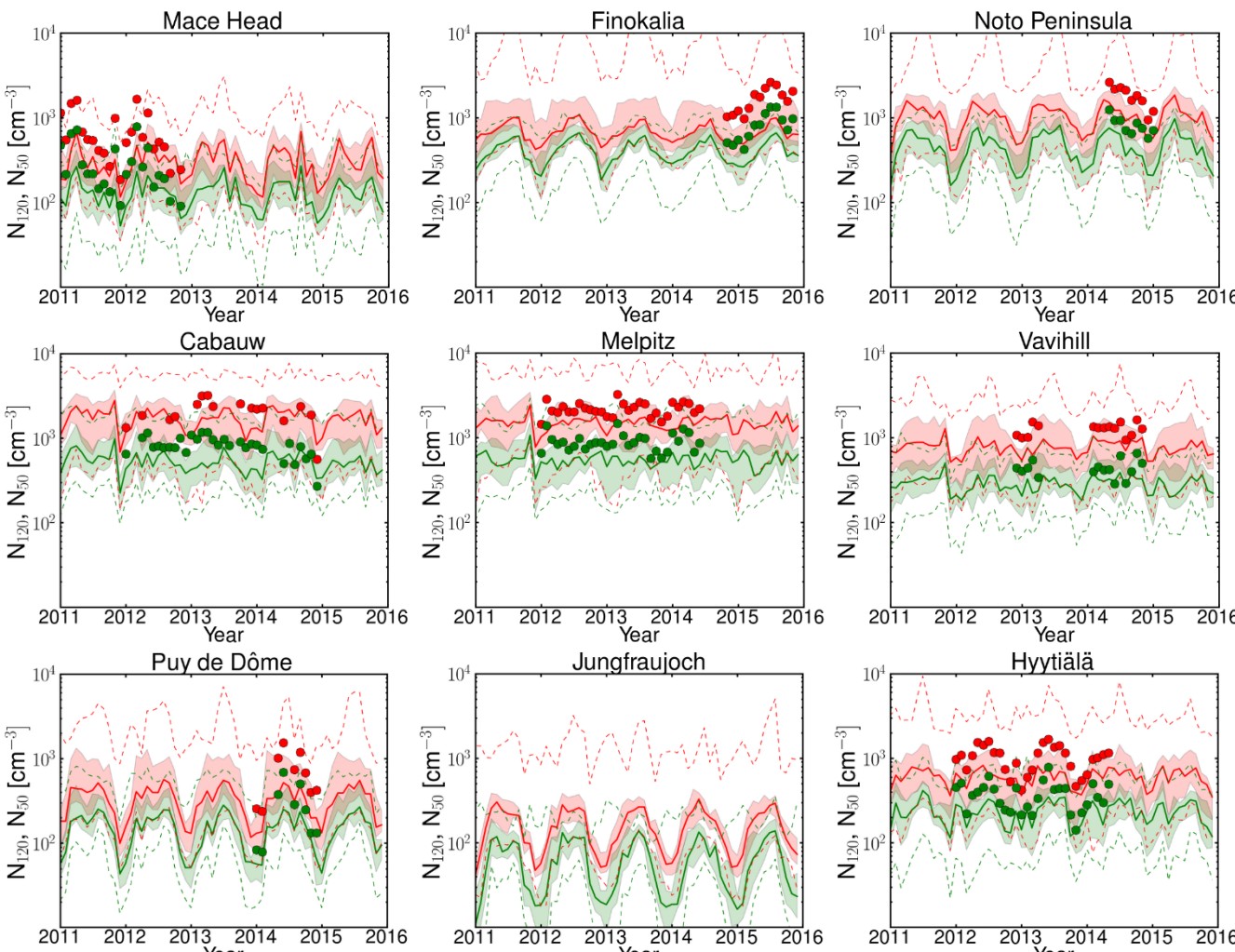

**Fig. 6.** Monthly ensembles for the period 2011-2015 of the number concentration of particles with diameters larger than 50 nm ($N_{50}$ – in red) and 120 nm ($N_{120}$ – in green). The continuous lines correspond to the median of the models for each month, the shaded areas show the 25/75% quartiles and the dashed lines the minimum and maximum of all models for the $N_{50}$ (red area) and $N_{120}$ (green area). Observational data are available for all stations except Jungfraujoch and are shown with symbols of the corresponding color.

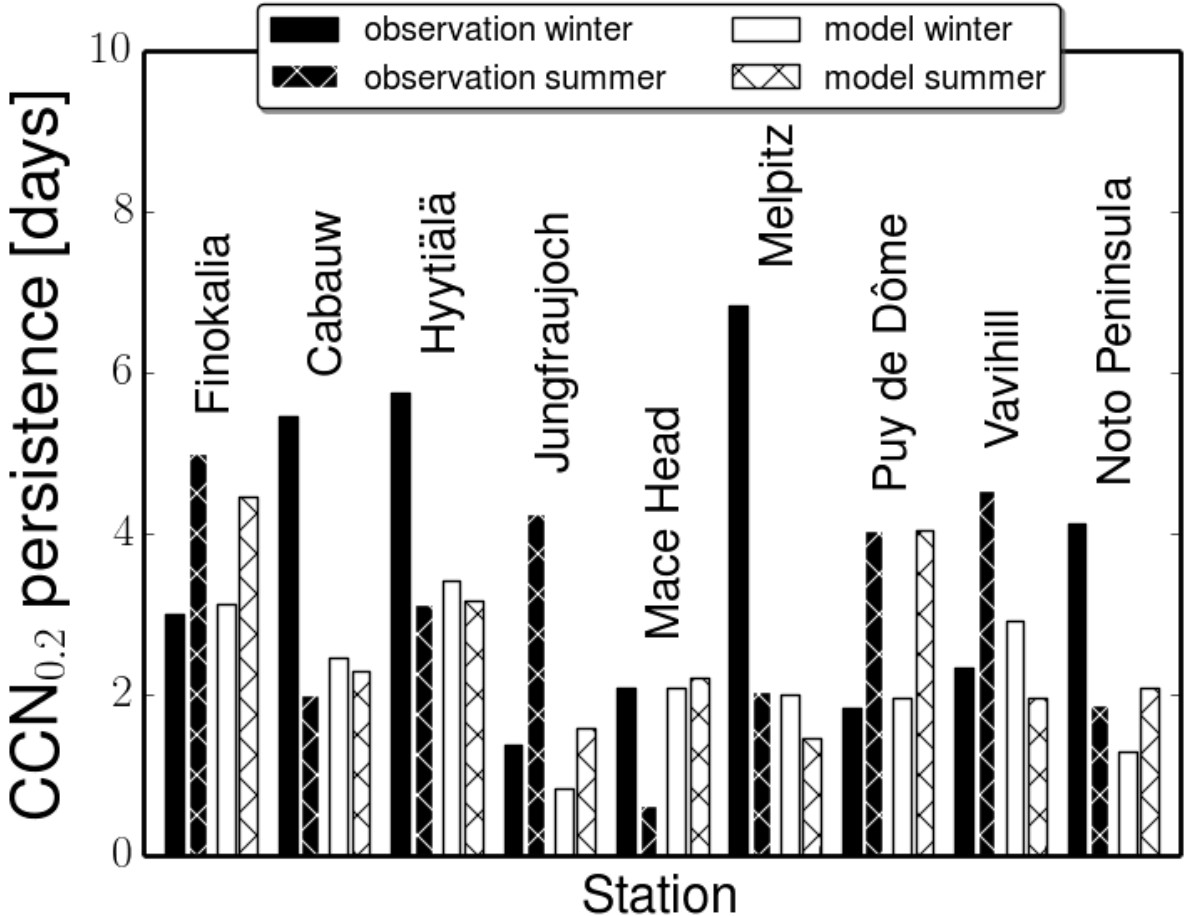

**Fig. 7.** Comparison between the observed and the mean of the model derived persistence (days) of $CCN_{0.2}$ during winter (left bar) and summer (right shaded bar) for each station. The observed persistence times are shown in black for each station and the mean of the model-derived persistence times in white. The persistence times obtained from model simulations have been computed at the same time periods as the observed ones.

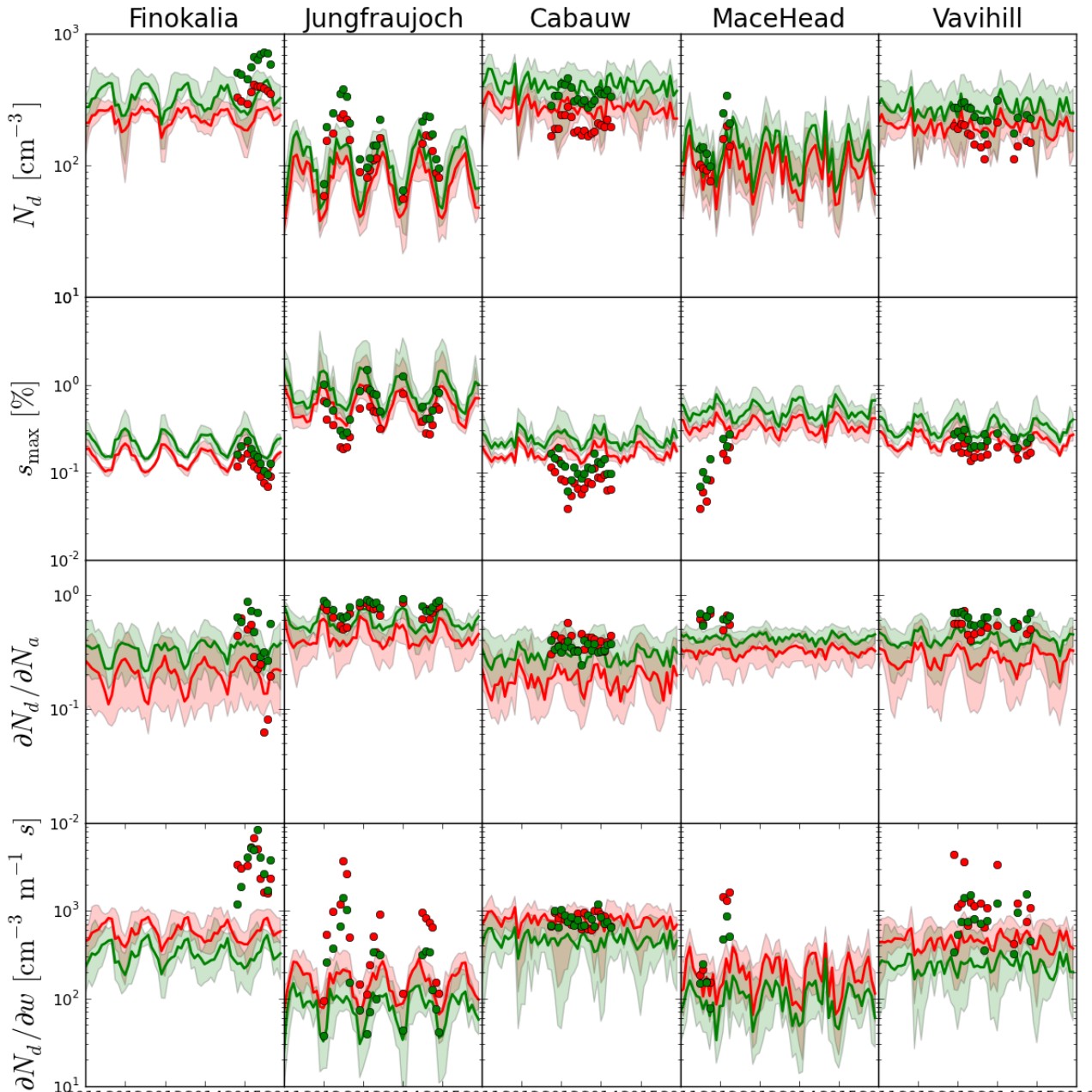

**Fig. 8.** Comparison between the observed (symbols) and the monthly averages of all models (continuous lines) of the cloud droplet properties; in red for updraft velocity w=0.3 ms[-1] and in green for updraft velocity w=0.6 ms[-1]. For each station from top to bottom the four graphs show (as indicated in the *y*-axis label), the number of cloud droplets, $N_d$, the maximum supersaturation, $s_{max}$, the sensitivity of the $N_d$ to the total number of aerosol particles, $(\partial N_d/\partial N_a)$, and the sensitivity of the $N_d$ to the updraft velocity $(\partial N_d/\partial w)$.

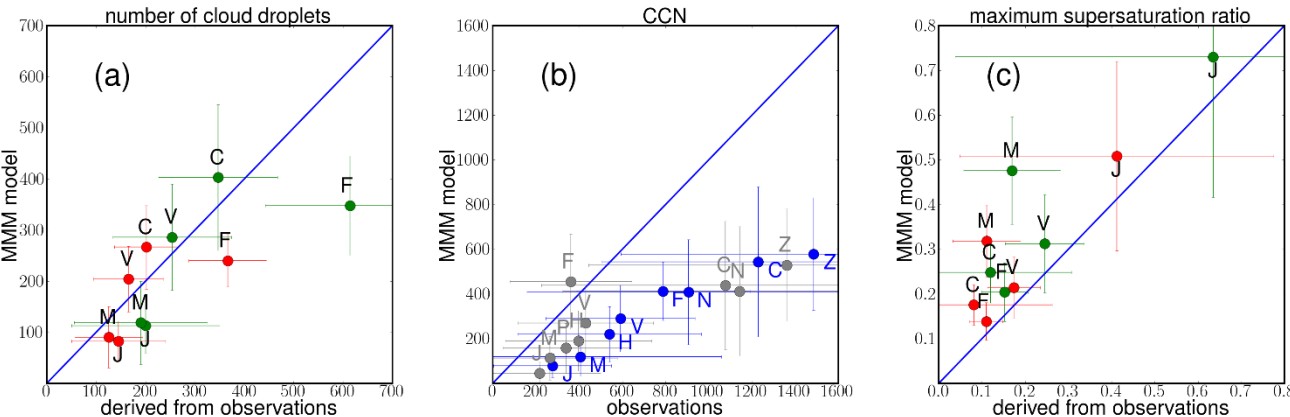

**Fig. 9.** Scatter plot of the average of multi-model median results (*y*-axis) versus observationally-derived results (*x*-axis) for :
(a) CDNC ($N_d$) (in cm$^{-3}$) in red for updraft velocity $w = 0.3\ ms^{-1}$ and in green for updraft velocity $w = 0.6\ ms^{-1}$, (b) CCN
at supersaturation 0.2% (gray) and CCN at maximum supersaturation (blue) with available data (in cm$^{-3}$). To fit the scale all
CCN number concentrations at maximum supersaturation (blue symbols) have been divided by 2. (c) as (a) but for s$_{max}$ (in
%). The letters close to the symbols indicate the station names (C - Cabauw, F - Finokalia, H - Hyytiälä, J - Jungfraujoch, M
- Mace Head, N - Noto Peninsula, P - Puy de Dôme, V - Vavihill, and Z - Melpitz)

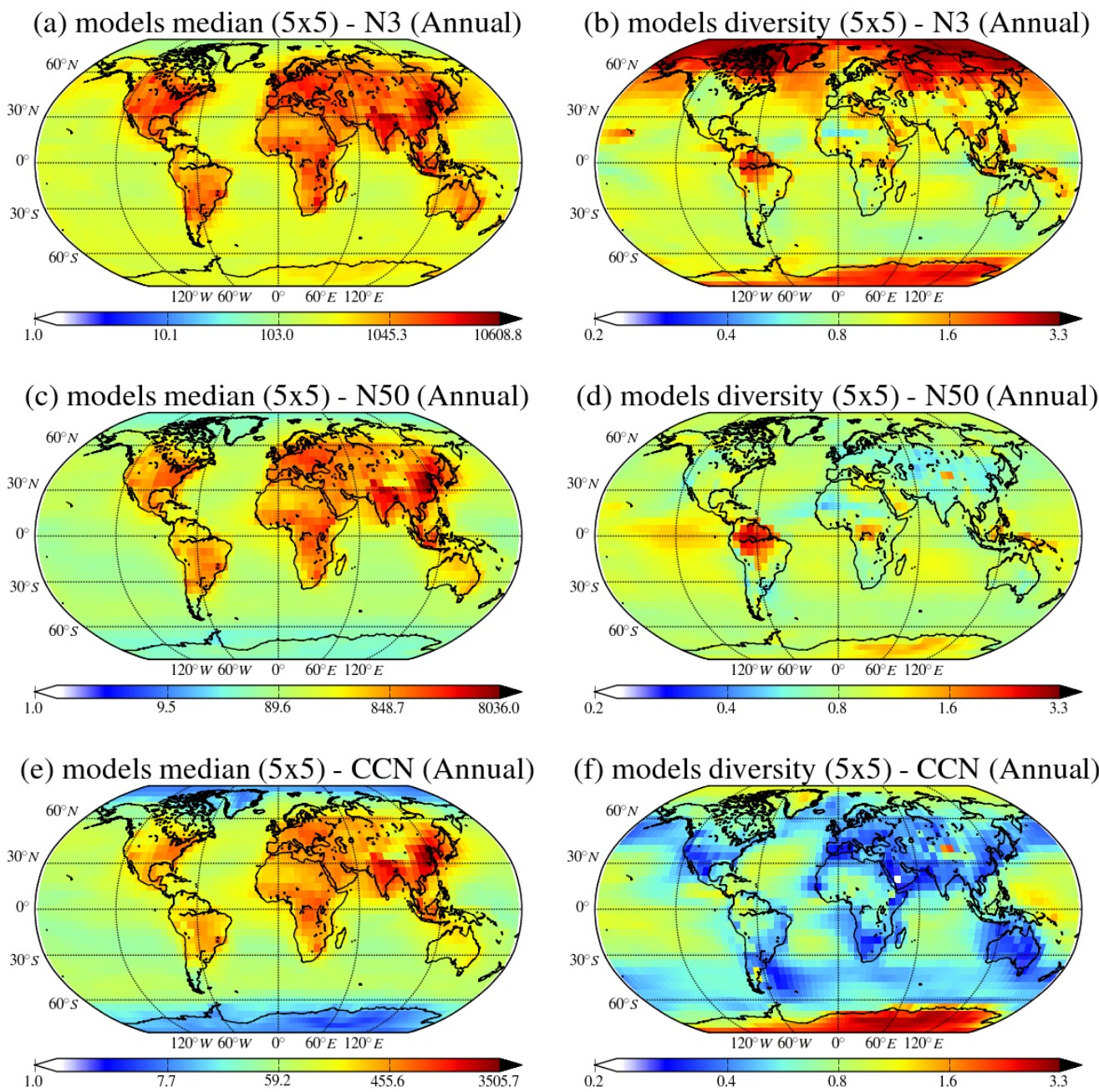

**Fig. 10.** Global distributions of the annual multi-model median concentrations of the $N_3$, $N_{50}$ and $CCN_{0.2}$ in cm$^{-3}$ for the year 2011 (a, c, e respectively) and the corresponding diversities (b, d, f, respectively; calculated as the ratio of standard deviation to the mean of the models).

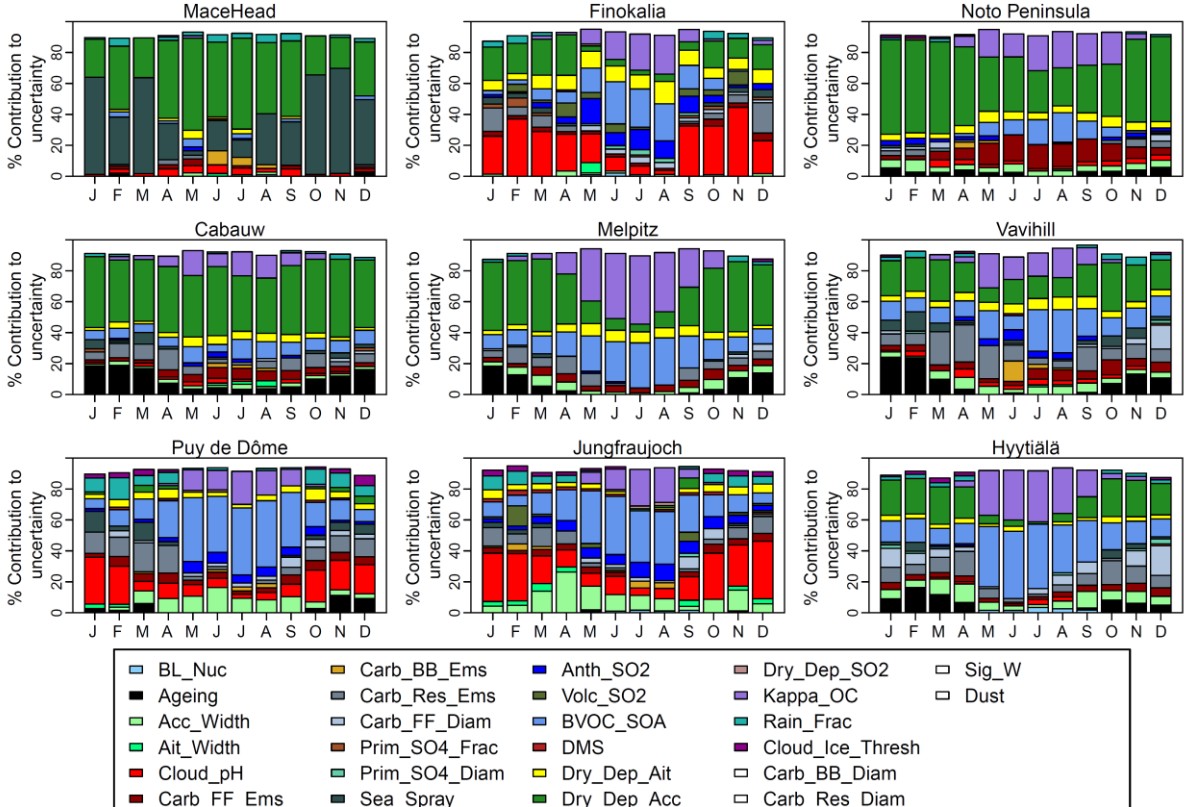

**Fig. 11.** Contribution to the uncertainty in monthly average $CCN_{0.2}$ based on HadGEM3-UKCA perturbed parameter ensemble simulations for year 2008. Each colour refers to one of the 26 perturbed parameters as indicated in the legend of the figure. The uncertainty is shown as the percentage contribution of the parameter to the $CCN_{0.2}$ variance. The assumed parameter uncertainty ranges are given in Yoshioka et al. (under review 2019). All contributions smaller than 1% are not shown.

Abbreviations are: BL_Nuc : Boundary layer nucleation; Ageing : Ageing "rate" from insoluble to soluble ; Acc_Width : Modal width (accumulation soluble/insoluble) ; Ait_Width  : Modal width (Aitken soluble/insoluble); Cloud_pH : pH of cloud drops; Carb_FF_Ems :  Particle mass emission rate for BC and OC (fossil fuel); Carb_BB_Ems : Particle mass emission rate for BC and OC (biomass burning); Carb_Res_Ems :  Particle mass emission rate for BC and OC (biofuel) ;Carb_FF_Diam : Particle emitted mode diameter for BC and OC (fossil fuel); Carb_BB_Diam : Particle emitted mode diameter for BC and OC (biomass burning) ; Carb_Res_Diam : Particle emitted mode diameter for BC and OC (biofuel); Prim_SO4_Frac : Mass fraction of $SO_2$ converted to new $SO_4^{-2}$ particles in sub-grid power plant plumes; Prim_SO4_Diam : Mode diameter of new sub-grid $SO_4^{-2}$ particles ; Sea_Spray : Sea spray mass flux (coarse/accumulation); Anth_SO2 : $SO_2$ emission flux (anthropogenic) ; Volc_SO2 : $SO_2$ emission flux (volcanic); BVOC_SOA : Biogenic monoterpene production of SOA; DMS : DMS emission flux; Dry_Dep_Ait : Dry deposition velocity of Aitken mode aerosol ; Dry_Dep_Acc : Dry deposition velocity of accumulation mode aerosol; Dry_Dep_SO2 : Dry deposition velocity of $SO_2$ ; Kappa_OC : Hygroscopicity parameter Kappa for organic aerosols. Default value in UKCA is 0.06. See Petters and Kreidenweis 2007 ACP; Sig_W : Standard deviation of updraft velocity. (This affects activation of aerosol particles to form cloud droplets.); Dust : dust emission flux; Rain_Frac : The fraction of the cloudy part of the gridbox where rain is forming and hence scavenging takes place; Cloud_Ice_Thresh : Scavenging (by both cloud liquid and ice water) is suppressed in dynamic clouds when cloud ice fraction is higher than this value.