# Peer review of "Evaluation of global simulations of aerosol particle and cloud condensation nuclei number, and implications for cloud droplet formation"

_Atmospheric Chemistry and Physics, 2018_

## Referee Comment (RC1) · Anonymous Referee #1 · 31 Jan 2019

This study presents results form a model intercomparison of 16 global models to compare predicted aerosol and cloud condensation nuclei number concentrations. Model results are compared to observations at nine locations representative for differently polluted regions. Based on predicted and observed CCN concentrations, cloud drop number concentrations are calculated for each model output and location. This intercomparison is accompanied by performing a perturbed parameter ensemble study that explores the sensitivities of model/observation biases to numerous aerosol parameters. This comprehensive study is extremely useful as it highlights uncertainties in current model predictions of aerosol parameters that translate into uncertainties in cloud drop number concentrations. Given the large uncertainties in current estimates of aerosol-

cloud-interactions, such studies will help to refine future models. The manuscript is extremely well written and I only have some minor comments that should be addressed prior to publication.

Main comments

In large parts, the discussion of the model results is merely a description of the figures. I missed at several places, a more thorough discussion of the underlying potential reasons of discrepancies and comparison to prior model studies. Examples are listed in the following:

p. 10, l. 32: The sensitivity of CCN prediction at low supersaturations has been discussed in many previous studies and been ascribed to the sensitivity of N(CCN) if the critical diameter is within the 'steep' section of the size distribution. Some references should be added here.

p. 13, 20-26: Is it expected that PPE simulations show this qualitative agreement with the MMM? What would it mean if there were no agreement?

p. 14, l. 5-7: Is this underprediction of total aerosol particle number a known bias in global models? Wouldn't it be then more useful to report CCN fractions, i.e. N(CCN)/ Na?

p. 15, l. 20: How do the parameterizations of wet deposition differ in the various models? I am quite certain that such descriptions have been discussed in previous model intercomparison and been identified as major causes of discrepancies. Such references should be added here. p. 16, l. 20-25: The relationship between N(CCN) and Nd as a function of w has been explored in many theiretical studies (e.g. by Nenes, Feingold, Reutter and others). Can your 'qualitative example' here be related to them?

p. 17, l. 20/21: What are the different assumptions on emission heights in the various models? How much of a discrepancy did the study by Daskalakis find?

p. 18, l. 11: Is there an explanation of the much highe rconcentraiton as found by

Spracklen et al.?

p. 21, l. 2: Your findings are in agreement with the concept of aerosol-'vs 'updraft-limited' regimes. That should be repeated here.

Minor comments

Title: I think the title should be reworded and 'number' should be moved: Evaluation of global simulations of aerosol particle and cloud condensation nuclei number, and implications for cloud droplet formation'

p. 4, l. 9: This sentence implies that particle size also determines hygroscopicity – which is not true as hygroscopicity is a mass-related parameter. Please reword.

p. 6, l. 14: 'one of the different binary homogenous nucleation' sounds weird. Please reword.

p. 6, l. 29-p. 7, l. 3: It might be easier to list this information in a table.

p. 8, l. 4, l. 9 and later: In the reference list, Schmale 2017a and 2017b are listed whereas in the text Schmale et al., 2017 and 2018 are cited. Please correct.

p. 8, l. 20: 'the chemical composition...': There seems to be a verb missing in this fragment.

p. 10, l. 16: why 'would no doubt affect'? I think it is safe to say that 'no doubt affects...'

p. 11, l. 29: higher summer time values...

p. 12, l. 27: I suggest rewording to 'an insufficient number of small particles are predicted to activate in the model'

p. 12, l. 29: remove 'the' (for all stations...)

p. 14, l. 19: Where does the parenthesis close?

p. 14, l. 19: remove 'the' (by nine of the models...)

p. 14, l. 19: I do not understand the fragment 'not systematically the same models at all stations'

p. 15, l. 2: Has 'persistence' been defined anywhere earlier? (I might have missed it). Is it the time a CCN spends in the atmosphere?

p. 18, l. 23/4: Given that the referenced figures are in supplement, the 'significant differences' should be more explicitly discussed here.

p. 19, l. 3-21: It would help a lot to connect this text to Figure 11, if the abbreviations from the figure legend are added to the text. Some of them are not intuitive.

p. 19, l. 18: 'in-cloud oxidation of SO2' is not explicitly included in the parameters in Figure 11. Is it connected to uncertainties in pH?

p. 26, l. 15 and l. 17: The second of these two references seems wrong and redundant.

Figures 2 and 6, caption: 'dots and dashed lines' and 'bold points and dashed lines', respectively, is confusing as the dashed lines are model results. I think it should be sufficient to say 'observational data is shown with symbols' (unless I misunderstand something here). Figure 7: It would be much easier to understand the figure if you added a legend with 'observation winter', 'observation summer', 'model winter', 'model summer'

Figure 8: Is Smax shown in %?

Figure 9, caption: 'blue colored values' sounds strange (values don't have colors). I suggest rewording to 'all NCCN at smax (blue symbols) have been divided by 2'

Figure 11: a) I understand that details about the analysis will be presented by Yoshioka et al. However, given that this paper is still in preparation, I suggest adding a list of all parameters, their meaning and ranges. b) Sum of the bars do not add up to 100%. Is the rest then the sum of the last four parameters, i.e. Carb_BB_diam, carb_res_diam, Sig_w, and Dust?

---

## Referee Comment (RC2) · Anonymous Referee #2 · 12 Feb 2019

Fanourgakis et al. perform a large model intercomparison study focusing on the evaluation of aerosol size distributions, CCN, and Nd with surface observations as a reference. The main conclusion is that models have substantial problems in simulating the concentrations of CCN, with on average strong underestimations. In contrast, Nd as derived applying an off-line parameterisation with prescribed vertical wind is much better correlated to observations-tied estimates.

The study is a very large effort and worth publishing. It is also useful for upcoming assessments such as the IPCC AR6. It is, however, a pity that it was impossible to point to specific parameterisation shortcomings, since obviously neither in the multi-model

ensemble nor in the PPE, specific parameterisation choices and/or specific models seemed to systematically correlate better to the observations than others. The only hint is that organic aerosols seem to be more difficult than other types.

I think the authors should better explore the Nd result in a revision. The first thing that would be very useful and probably not difficult to do is to compare the Nd the models actually compute to the idealised ones computed here off-line. It would be very interesting to see how large the potential biases in the parameterisations of droplet activation are in comparison to the aerosol biases. I also wonder why the deviations in Nd are in some cases qualitatively different from the deviations in CCN. Some explanation is needed.

Specific comments

p4 l12: Is "OA" really defined as a fraction here? - contradiction to p8 l8

p5 l13: really opposite, i.e. of different sign?

p6 l30: Modal aerosol module

p7 l2: Kirkevåg

p7 l11: "a few models account for melting and sublimation of ice crystals" - I have the impression this sentence is incomplete.

p10 l15: why this very coarse resolution, and not just the resolution of the coarsest-resolved model?

p11 l6: the conclusion that the correlation is captured "satisfactorily" needs quantification: what can be considered "satisfactory"? How do the authors quantify these relative differences? - perhaps a figure like Fig. 3 but for the spatial variability would be useful?

p11 l7: The index of agreement, since it is not a conventional metric, would need to be defined in the main text. The authors further should motivate why this IoA provides extra information that substantially goes beyond NMB and NME.

p13 l1: "satisfactorily" - again a quantification would be helpful. Is this significantly better than for CCN0.2?

p14 l5: the + for NME is unnecessary

p14 l10-12: Why would the authors call an NME of 97% "reasonable"?

p15 l6: I rather have the impression that it is completely unclear: in 5/9 station, it is indeed longer in winter, but in 4/9 the opposite.

p15 l7: in 6/9 cases, the MMM is consistent with the summer-winter change as the obs show, in 3/9 cases, opposite. Is that "qualitatively capturing" the change?

p15 l32: it doesn't hinder an eventual decrease in smax, it implies it.

p16 l24: The competition effect cannot explain why the ratio observed/simulated for Cabauw and Vavihill turns from substantial overestimation to underestimation, and why the opposite is found for Finokalia.

p31 Fig. 2 – I don't see the dashed lines for the observations.

p34 Fig. 5 – Would it be useful to show the MMM as well?

---

## Author Comment (AC1) · 27 Apr 2019

Reply to Reviewer comments: Anonymous referee 1

*This comprehensive study is extremely useful as it highlights uncertainties in current model predictions of aerosol parameters that translate into uncertainties in cloud drop number concentrations. The manuscript is extremely well written and I only have some minor comments that should be addressed prior to publication. In large parts, the discussion of the model results is merely a description of the figures. I missed at several places, a more thorough discussion of the underlying potential reasons of discrepancies and comparison to prior model studies. Examples are listed in the following:*

Reply: We thank the reviewer for the positive comments and the very careful reading of the manuscript. We have now taken all of them into account in the revised version of the manuscript. More thorough discussion of the results have been added in the revised version of the manuscript. We provide here below a point-by-point reply to the specific comments.

*p. 10, l. 32: The sensitivity of CCN prediction at low supersaturations has been discussed in many previous studies and been ascribed to the sensitivity of N(CCN) if the critical diameter is within the 'steep' section of the size distribution. Some references should be added here.*

Reply: As suggested we now refer to a few relevant references with appropriate comments :

'Such hypothesis is supported by earlier studies that have observed large size-dependence of the sensitivity in activation fraction at low supersaturations and in the size ranges between 60 and 100 nm (Bougiatioti et al., 2011). Deng et al. (2013) reported inferred critical diameters varying by factors of 2-3 for low supersaturations from 0.2% to 0.06% and suggested the use of size-resolved particle number concentrations with inferred critical diameters or size-resolved activation ratios to predict CCN. Errors in CCN predictions have been shown to exceed 50% only at very low supersaturations (Reutter et al., 2009) and reach a factor of 2.4, while at high supersaturations CCN overestimate can be less than 5% (Ervens et al., 2007). The global near-surface mean CCN prediction error has been estimated at about 9% and regionally the maximum error can reach 40% (Sotiropoulou et al., 2007). The largest CCN prediction error was found in regions with low in-cloud $s_{max}$, like those affected by long-range transport of pollution or industrial pollution plumes, and the lower CCN prediction error in regions where in-cloud $s_{max}$ is high, which is typical for pristine areas. Sotiropoulou et al. (2007) also found that the assumption of size-invariant chemical composition of internally mixed aerosol increases the error by a factor of two.'

*p. 13, 20-26: Is it expected that PPE simulations show this qualitative agreement with the MMM? What would it mean if there were no agreement?*

Reply: As explained in the last paragraph of section 2.1, PPE simulations consist of 235 simulations covering the most probable ranges of 26 parameters controlling aerosol processes and emissions perturbed simultaneously. Therefore it is expected that a combination of some of the PPE members will be very similar to MMM members and consequently that the PPE will qualitatively agree with the MMM, enabling the use of PPE results to identify some potential causes of uncertainty in CCN calculations compared to observations in section 5. If there was no agreement, it would mean that there are more parameters or processes that dominate the aerosol processes and/or emissions that are not considered in the PPE and therefore such an analysis would not have been possible.

A corresponding comment will be added at the end of section 3.2. 'The qualitative agreement between PPE and MMM indicates that the perturbed parameters are those having a significant control on aerosol processes and emissions and can be used for CCN uncertainty attribution in section 5.'

*p. 14, l. 5-7: Is this under prediction of total aerosol particle number a known bias in global models? Wouldn't it be then more useful to report CCN fractions, i.e. N(CCN)/Na?*

Reply: Indeed, an earlier AEROCOM intercomparison of 12 global models by Mann et al. (2014) revealed a systematic underprediction across models in the number concentration of accumulation sizes that are relevant to CCN.

Following the reviewer's suggestion, we have calculated the activation ratios for $N_{120}$ ($CCN_{0.2}$ to $N_{120}$), $N_{80}$ ($CCN_{0.2}$ to $N_{80}$) and for $N_{50}$ ($CCN_{0.2}$ to $N_{50}$) particles. None of these ratios shows consistent behavior between the models and observations at all stations. At some stations (e.g., Hyytiala) the modeled activation fraction seems to agree with the observations within the observed variability, while at others (e.g. Melpitz, Finokalia and Noto Peninsula) it is totally different. This result supports the understanding that the aerosol number concentration cannot be used as a proxy for CCN levels since no fixed critical diameter can be specified at which an aerosol particle becomes a CCN. Activation of aerosols to CCN depends on the size distribution and the chemical composition of the aerosols as well as on the supersaturation that develops in clouds (e.g., Seinfeld and Pandis, 2006; Kalkavouras et al., ACPD 2018). As an example we show below the calculated activation fractions of N120 for Hyytijala, Finokalia, Melpitz and Noto Peninsula.

For clarity we added the following sentence in the revised version. 'Note however that the aerosol number concentration cannot be used as a proxy for CCN levels since activation of aerosols to CCN depends not only on the size distribution but also on the chemical composition of the aerosols as well as on the supersaturation that develops in clouds (e.g., Seinfeld and Pandis, 2006; Kalkavouras et al., 2018).

[Figure]

[Figure]

[Figure]

[Figure]

Figure: Activation fractions from the models (red lines) and from the observations (black lines) calculated as the ratio of CCN at 0.2% supersaturation to $N_{120}$ (defined as the number concentration of aerosol particles having diameter larger than 120 nm).

*p. 15, l. 20: How do the parameterizations of wet deposition differ in the various models? I am quite certain that such descriptions have been discussed in previous model intercomparison and been identified as major causes of discrepancies. Such references should be added here.*

Reply: We now added a new table in the supplement (Table S4) with information on the parameterizations used in the models for dry and wet deposition in section 2.1 and we modified and extended this part of the discussion in section 3.4 as follows:

In section 2.1: 'Both dry deposition and wet deposition of aerosol particles are taken into account in the participating models as shown in the supplementary Table S4. For the dry deposition models account for gravitational settling and for turbulence, thus these processes depend on the aerosol particle size. The omission of super-coarse particle sources associated with dust and sea-salt particles result in discrepancies between models, and, between model results and observations (Myriokefalitakis et al., 2016). Wet deposition parameterizations account for both in-cloud scavenging, which is sensitive to the solubility of aerosol particles, and below-cloud scavenging by convective and large-scale precipitation (Seinfeld and Pandis, 2006).'

In section 3.4: "In addition to atmospheric transport patterns, dry and wet deposition processes are presumably affecting the persistence time. Because the present exercise was not focusing on deposition of aerosols, it does not have the necessary elements to elaborate on differences in the results associated with differences in the deposition parameterizations. However, earlier global model comparisons provide insight to such differences. Tsigaridis et al. (2014) comparison of thirty-one global models among which those participating in the present study has shown that the representation of aerosol microphysics in the models was important for dry deposition. In particular, they have shown that the use of M7 aerosol microphysics module was associated with low dry deposition fluxes of organic aerosol, which is mainly fine aerosol in the models, and the dry deposition rate coefficient ranged from 0.005 to 0.13 day$^{-1}$, i.e. with a max/min ratio of 26. They also found that the effective wet

deposition rate coefficient in the 31 participating models ranged from 0.09 to 0.24 day$^{-1}$, i.e. with a max/min ratio of 2.6 that is 10 times lower than for dry deposition, and found virtually no change between AEROCOM phase I and AEROCOM phase II models. Kim et al. (2014) compared the deposition of dust, which is mainly coarse aerosol, calculated by a smaller subset (5) of AEROCOM models. They pointed out that the size distribution of dust differs among these models and found a 30% difference in the effective dry deposition rate coefficient and about the same in the total deposition rate varying from 0.28 to 0.37 day$^{-1}$. Kim et al (2014) analysis also revealed differences in the annual precipitation rate and in its seasonal distribution in the models and a factor of 2 differences in the fraction of wet to the total deposition of dust among the models (ranging between 0.36 and 0.63). In addition the PPE results (see section 5) clearly show that dry deposition is one of the major factors of uncertainty in the calculations of CCN at 0.2% supersaturation. Kristiansen et al. (2016) investigated the causes of differences in aerosol lifetimes within 19 global models by making use of an observational constraint from radionuclide measurements and found largely underestimated accumulation-mode aerosol lifetimes due to too fast removal in most models. In particular, they found that the way aerosols are transported and scavenged in convective updrafts makes a large difference in aerosol vertical distribution and lifetimes, as revealed in their simulations from the same model (CAM5) but with different convective transport and wet removal treatments (Wang et al., 2013)".

*p. 16, l. 20-25: The relationship between N(CCN)and Nd as a function of w has been explored in many their etical studies (e.g. by Nenes, Feingold, Reutter and others). Can your 'qualitative example' here be related to them?*

Reply: This observed behavior is indeed based on the relationship between N(CCN) and $N_d$, but most studies discuss the slope ($\partial N_d/\partial CCN$), evaluated at particular model states or the relative importance of input parameters that drive $N_d$ variability. Very few studies actually discuss within the context of error propagation from CCN model-observation discrepancy to the implied cloud droplet number uncertainty, and none with the type of in-situ data (many locations, long time series) available here. In the introduction (page 5, paragraphs 1,2), relevant references are provided; in the revised manuscript, the aforementioned section will be referred to, we will also cite a few more manuscripts that directly discuss the issue of CCN error propagation to droplet number.

'In agreement with our finding, Sotiropoulou et al. (2006) using a similar approach applied to observations from the ICARTT field campaign estimated that a 20–50% error in CCN closure results in a 10–25% error in $N_d$, while global simulations suggest global average CCN prediction error between 10 and 20% and a smaller corresponding $N_d$ error between 7 and 14% (Sotiropoulou et al., 2007). Such reduction in error can be explained by a self-regulation by $N_d$ since $S_{max}$ decreases with increasing aerosol number concentration as discussed by many studies published to date (e.g., Twomey et al., 1959; Charlson et al., 2001; Nenes and Seinfeld, 2003; Feingold and Siebert, 2009), giving rise to regions where $N_d$ is relatively insensitive to changes in CCN or updraft velocity (e.g., Rissman et al., 2004, and Reutter et al., 2017). At very high CCN levels, and in the presence of sufficient large hygroscopic CCN, $N_d$ may actually decrease with increases in aerosol amount (Ghan et al., 1998; Feingold, 2001; McFiggans et al., 2006, Reuter et al., 2017); parameterizations that do not fully capture these important aspects of the aerosol-droplet relationship may also give rise to biases in aerosol indirect forcing assessments (e.g., Morales-Betancourt et al., 2014a).'

*p. 17, l. 20/21: What are the different assumptions on emission heights in the various models?*
*How much of a discrepancy did the study by Daskalakis find?*

Reply: In the revised version we will provide a short discussion on the assumptions of biomass burning emission heights in the participating models.

Most of the participating models (new supplementary Table S4) follow the AEROCOM recommendation of biomass burning emission heights which in the boreal regions extend above 2 km and up to 6 km for the Canadian boreal fires (Dentener et al., 2006). ECHAM6-HAM2 and ECHAM6-HAM2-AP use a slightly different vertical distribution of biomass burning emissions with 75% within the planetary boundary layer (PBL), 17% in the first and 8% in the second level above the PBL (Tegen et al., 2018). EMAC assumes biomass burning emissions at 140 m and GEOS-Chem-APM well mixed in the boundary layer. This information is now added in section 2.1.

Furthermore, Daskalakis et al. (2015) found that the calculated tropospheric lifetimes of pollutants can differ by up to 30% when injected at heights following Dentener et al. (2006) compared to be emitted solely in the lowest model layer. Emitting aerosols in the free troposphere reduces their concentrations near surface and increases them in the middle troposphere; the largest differences of about 25% in the tropospheric column were calculated for the high latitudes over North America and China. Furthermore, Jian and Fu (2014) calculated 50–150% more BC at 700 hPa than when emitted in the boundary layer.

We rephrase the relevant sentence to provide relevant information in the revised manuscript: 'Assumption of the injection height is also a source of discrepancy between models, leading to differences in the calculated lifetimes (up to 30%) and in the tropospheric columns (up to 25%) of pollutants (Daskalakis et al., 2015), while differences of an order of magnitude in their concentrations are computed for the middle troposphere (Jian and Fu, 2014). Thus, differences in the emission injection heights in the participating models, as outlined in section 2.1 and Table S4, contribute to the model results divergence'.

*p. 18, l. 11: Is there an explanation of the much higher concentration as found by Spracklen et al.?*
Reply: To be more precise in Spracklen et al (2011) paper (Figure 1c) CCN computed accounting for carbonaceous aerosols are in the range of 3,162-10,000 cm-3 and these very high values are computed over China and attributed to carbonaceous aerosols acting as CCN.

For clarity in the revised version we rephrase this sentence accordingly. 'This value is in the range of the 3,162-10,000 cm$^{-3}$ CCN$_{0.2}$ concentrations simulated by Spracklen et al. (2011) over China and attributed to carbonaceous aerosols acting as CCN.'

*p. 21, l. 2: Your findings are in agreement with the concept of aerosol-'vs 'updraft-limited' regimes. That should be repeated here.*

Reply: This comment has also been raised above. Please see our related response; a similar reference is made at this point in the text as well:
"and support the concept of existence of two distinct regimes the aerosol-limited and the updraft-limited." Is added at the end of the sentence.

**Minor comments**

*Title: I think the title should be reworded and 'number' should be moved: Evaluation of global simulations of aerosol particle and cloud condensation nuclei number, and implications for cloud droplet formation'*

Reply: We moved 'number' as suggested

*p. 4, l. 9: This sentence implies that particle size also determines hygroscopicity –which is not true as hygroscopicity is a mass-related parameter. Please reword.*

Reply: We thank the reviewer for pointing out this misleading sentence, we rephrased as following: 'Although less important than particle size for CCN formation, particle chemical composition determines aerosol hygroscopicity'.

*p. 6, l. 14: 'one of the different binary homogenous nucleation' sounds weird. Please reword.*

Reply: We removed 'one of the different'

*p. 6, l. 29-p. 7, l. 3: It might be easier to list this information in a table.*

Reply: This table exists in the supplement (Table S2). The text is meant to be a brief outline of the discussion in S1 and the table S2 in the supplement. In the revised version we explicitly direct the reader to supplementary table S2:
'Supplementary tables S1, S2, S3, and S4 provide a summary of the main features of the participating model. Further details are provided in these tables.'

*p. 8, l. 4, l. 9 and later: In the reference list, Schmale 2017a and 2017b are listed whereas in the text Schmale et al., 2017 and 2018 are cited. Please correct.*

Reply: We thank the reviewer for pointing out this discrepancy. We updated the reference list to use include the finally published paper Schmale et al., 2018 in Atmos. Chem. Phys.

*p. 8, l. 20: 'the chemical composition...': There seems to be a verb missing in this fragment.*

Reply: We replaced the semi-column by 'as well as'.

*p. 10, l. 16: why 'would no doubt affect'? I think it is safe to say that 'no doubt affects...'*

Reply: It is now corrected as suggested.

*p. 11, l. 29: higher summer time values...*

Reply: done

*p. 12, l. 27: I suggest rewording to 'an insufficient number of small particles are predicted to activate in the model'*

Reply: this is modified as suggested by the reviewer

*p. 12, l. 29: remove 'the' (for all stations...)*

Reply: done

*p. 14, l. 19: Where does the parenthesis close?*

Reply: The parenthesis closes after the %

*p. 14, l. 19: remove 'the' (by nine of the models…)*

Reply: done

*p. 14, l. 19: I do not understand the fragment 'not systematically the same models at all stations'*

Reply: We mean that there are no specific models that systematically perform better than others, as we explained in the sentence that was following ' Because different models are appearing as outliers at each station…' To avoid confusion we removed the fragment in question.

*p. 15, l. 2: Has 'persistence' been defined anywhere earlier? (I might have missed it).Is it the time a CCN spends in the atmosphere?*

Reply: In page 9, line 2 section 2.4 subsection 'CCN persistence' we define it as 'the duration for which the CCN number concentrations remains similar to its earlier concentration, the autocorrelation …' as we direct the reader to the publication by Schmale et al 2018.

For clarity that sentence in section 2.4 now reads as follows: 'To investigate the duration for which the CCN number concentrations remains similar to its earlier concentration, the so-called persistence, the autocorrelation…'

*p. 18, l. 23/4: Given that the referenced figures are in supplement, the 'significant differences' should be more explicitly discussed here.*

Reply: Following the reviewer's suggestion appropriate discussion has been added: ' In particular, for all models near-surface BC distributions maximize over China, while individual models differ by a factor of 3 to 4. Simulated SS distributions maximize over the southern oceans where the models show the largest differences up to 2 orders of magnitude reflecting large differences in the parameterized emissions of SS (see also supplementary Table S3). Finally, DU distributions show the largest spread among models with near surface values that differ up to a factor of 40.'

*p. 19, l. 3-21: It would help a lot to connect this text to Figure 11, if the abbreviations from the figure legend are added to the text. Some of them are not intuitive.*

Reply: The abbreviations will be explained in the Figure caption for clarity:

'Abbreviations are: BL_Nuc : Boundary layer nucleation; Ageing : Ageing "rate" from insoluble to soluble ; Acc_Width : Modal width (accumulation soluble/insoluble) ; Ait_Width  : Modal width (Aitken soluble/insoluble); Cloud_pH : pH of cloud drops; Carb_FF_Ems :  Particle mass emission rate for BC and OC (fossil fuel); Carb_BB_Ems : Particle mass emission rate for BC and OC (biomass burning); Carb_Res_Ems :  Particle mass emission rate for BC and OC (biofuel) ;Carb_FF_Diam : Particle emitted mode diameter for BC and OC (fossil fuel); Carb_BB_Diam : Particle emitted mode diameter for BC and OC (biomass burning) ; Carb_Res_Diam : Particle emitted mode diameter for BC and OC (biofuel);

Prim_SO4_Frac : Mass fraction of $SO_2$ converted to new $SO_4^{-2}$ particles in sub-grid power plant plumes; Prim_SO4_Diam : Mode diameter of new sub-grid $SO_4^{-2}$ particles ; Sea_Spray : Sea spray mass flux (coarse/accumulation); Anth_SO2 : $SO_2$ emission flux (anthropogenic) ; Volc_SO2 : $SO_2$ emission flux (volcanic); BVOC_SOA : Biogenic monoterpene production of SOA; DMS : DMS emission flux; Dry_Dep_Ait : Dry deposition velocity of Aitken mode aerosol ; Dry_Dep_Acc : Dry deposition velocity of accumulation mode aerosol; Dry_Dep_SO2 : Dry deposition velocity of $SO_2$ ; Kappa_OC : Hygroscopicity parameter Kappa for organic aerosols. Default value in UKCA is 0.06. See Petters and Kreidenweis 2007 ACP; Sig_W : Standard deviation of updraft velocity. (This affects activation of aerosol particles to form cloud droplets.); Dust : dust emission flux; Rain_Frac : The fraction of the cloudy part of the gridbox where rain is forming and hence scavenging takes place; Cloud_Ice_Thresh : Scavenging (by both cloud liquid and ice water) is suppressed in dynamic clouds when cloud ice fraction is higher than this value.'

*p. 19, l. 18: 'in-cloud oxidation of SO2' is not explicitly included in the parameters in Figure 11. Is it connected to uncertainties in pH?*

Reply: Yes this is connected to uncertainties in cloud pH. It is now clarified in the text.

*p. 26, l. 15 and l. 17: The second of these two references seems wrong and redundant.*

Reply: The reference has been removed

*Figures 2 and 6, caption: 'dots and dashed lines' and 'bold points and dashed lines', respectively, is confusing as the dashed lines are model results. I think it should be sufficient to say 'observational data is shown with symbols' (unless I misunderstand something here).*

Reply: We thank the reviewer for this suggestion and we have changed the caption accordingly.

*Figure 7: It would be much easier to understand the figure if you added a legend with 'observation winter', 'observation summer', 'model winter', 'model summer'*

Reply: In the revised version we added the proposed legend.

*Figure 8: Is Smax shown in %?*

Reply: Yes this is now specified in the axis title and in the caption.

*Figure 9, caption: 'blue colored values' sounds strange (values don't have colors). I suggest rewording to 'all NCCN at smax (blue symbols) have been divided by 2'*

Reply: The caption has been rephrased accordingly.

*Figure 11: a) I understand that details about the analysis will be presented by Yoshioka et al. However, given that this paper is still in preparation, I suggest adding a list of all parameters, their meaning and ranges. b) Sum of the bars do not add up to 100%. Is the rest then the sum of the last four parameters, i.e. Carb_BB_diam, carb_res_diam,Sig_w, and Dust?*

Reply:

a) The paper by *Yoshioka et al.* is now under review.  We have added this reference in the manuscript and we also added in the figure caption the meaning of the parameters as provided earlier.

b) Furthermore, figure 11 only shows individual parameter contributions to uncertainty greater than or equal to 1%. All contributions smaller than 1% are not shown (this is now clearly stated in the figure caption).  These may be other individual parameter contributions as well as joint interaction terms.  The difference between the height of the bars and 100% corresponds to these terms.  This also means that not all parameters (colors) in the legend will appear in every column.  The parameters carb_BB_diam, carb_res_diam, Sig_w, and Dust have no color as these parameters do not contribute to any column in any of the plots (there are no cases where the contribution to uncertainty from these parameters is >=1%).

---

## Author Comment (AC2) · 27 Apr 2019

Reply to reviewer #2 comments

*Fanourgakis et al. perform a large model intercomparison study focusing on the evaluation of aerosol size distributions, CCN, and Nd with surface observations as a reference. The main conclusion is that models have substantial problems in simulating the concentrations of CCN, with on average strong underestimations. In contrast, Nd as derived applying an off-line parameterisation with prescribed vertical wind is much better correlated to observations-tied estimates.*

*The study is a very large effort and worth publishing. It is also useful for upcoming assessments such as the IPCC AR6. It is, however, a pity that it was impossible to point to specific parameterisation shortcomings, since obviously neither in the multi-model ensemble nor in the PPE, specific parameterisation choices and/or specific models seemed to systematically correlate better to the observations than others. The only hint is that organic aerosols seem to be more difficult than other types.*
*I think the authors should better explore the Nd result in a revision.*

Reply: We thank the reviewer for the positive evaluation and we will improve the manuscript according the reviewers suggestions.

*The first thing that would be very useful and probably not difficult to do is to compare the Nd the models actually compute to the idealised ones computed here off-line. It would be very interesting to see how large the potential biases in the parameterisations of droplet activation are in comparison to the aerosol biases.*

Reply: We agree with the reviewer on the interest of evaluating potential biases in the parameterizations of droplet activation used in the models. We see this as a natural follow-up to the present work, but requires considerable additional work to diagnose and provide conclusive statements on – and careful design to get the most useful information. Towards that, we feel that our prior studies (Morales-Betancourt et al., 2014a) provide a good example, where the source of Nd prediction discrepancy for two state-of-the-art parameterizations in the CAM5 global model was unraveled using adjoint sensitivity analysis. The findings of that study, not only lead to identification of parameterization biases, but also pointed to exactly which aspects of the parameterization (i.e., water uptake from large CCN) were not captured adequately, leading to the highly improved droplet parameterizations (Morales-Betancourt and Nenes, 2014b) that was used in the current study.

To clarify this, a relevant comment has been added at the end of the first paragraph of the description of CDNC calculations in section 2.4.: 'Note that evaluation of the differences in CDNC calculations by the different models that are derived both from the parameterizations used and from their input variables would require a different model intercomparison design than here and is planned for the future. Morales-Betancourt et al. (2014a) provide a good example, where the source of Nd prediction discrepancy for two state-of-the-art parameterizations in the CAM5 global model was unraveled using adjoint sensitivity analysis. That study pointed to exactly which aspects of the parameterization (i.e., water uptake from large CCN) were not captured adequately, leading to the highly improved droplet parameterizations (Morales-Betancourt and Nenes, 2014b) that was used in the current study.'

*I also wonder why the deviations in Nd are in some cases qualitatively different from the deviations in CCN. Some explanation is needed.*

Reply: Indeed Finokalia and Cabauw provide $N_d$ calculations (Figure 8 and related sensitivities) that qualitatively differ from the deviations in CCN0.2 (CCN at 0.2% ss; Figure 2). We explain this pattern by the inability for models to fully capture (overestimate or underestimate) the levels of the largest particles (> 250nm) and/or their hygroscopicity. This is an important conclusion that will be emphasized in this study. These results demonstrate that CCN at selected supersaturations (and respective model-observation discrepancies) cannot be generally used as proxy for cloud droplet number behavior, since the maximum supersaturation that develops inside the cloud (hence droplet number) responds to changes in aerosol and vertical velocity levels. This is even further complicated by the potential for model biases to change even sign across at cloud-relevant supersaturations.

We extended the discussion as follows:
'Based on the behavior described above, one can explain the $N_d$ predicted from simulated and observed CCN spectra. This is straightforward for Jungfraujoch and Mace Head stations. For Cabauw and Vavihill the observed-to-simulated ratio turns from a substantial overestimation in $CCN_{0.2}$ to an underestimation in $N_d$, and the opposite is found for Finokalia. This can be explained as follows. At both Cabauw and Finokalia, $s_{max}$ derived from observations is very low (approaching in the summer 0.07% at Finokalia and 0.04% at Cabauw; Fig. 8). The models overestimate these low values of $s_{max}$ and such values are indicative of the presence of large particles (>250nm) with sufficient hygroscopicity at these stations that are not captured by the models. Indeed, at Cabauw the available observations of CCN at 0.1% supersaturation show a larger underestimate by the models than for $CCN_{1.0}$ and $CCN_{0.2}$ (Figure below-new supplementary figure S16), also pointing to a model underestimate of the largest particles (>250nm) which induce the very low $s_{max}$. The overestimate in $s_{max}$ leads to an underestimate in $N_d$ by the models for all seasons except winter at Cabauw when the models at high updraft velocity capture the observationally-derived $N_d$ levels. Furthermore at Finokalia, $CCN_{1.0}$ is underestimated year-around, indicating that, in addition to the largest particles, the very small particles (smaller than 50 nm) that activate at 1.0% supersaturation and/or their hygroscopicity are also underestimated by the models there. On the other hand, larger particles than 120 nm that activate at 0.2% supersaturation are overestimated especially in winter and slightly underestimated in summer. Therefore the global models have significant difficulties in capturing the aerosol size distribution and hygroscopicity at Finokalia – which in turn translate to counterintuitive discrepancies in $N_d$.
At Vavihill a somehow different behavior is found; the underestimate of CCN at supersaturations of 0.2% and 0.7% changes to an overestimate at supersaturation 0.1% mainly in summer (supplementary Figure S16), indicating an underestimate of fine particles and/or their hygroscopicity and an overestimate of the largest particles and/or their hygroscopicity in particular during summer. This agreement of model results with observations during winter and the overestimate of CCN at 0.1% supersaturation during summer can explain the similar behavior of modelled $N_d$
The difference between model and observationally-derived $\partial Nd/\partial w$, clearly supports the above statements. Since observations predict a suppressed $s_{max}$ compared to model distributions (Fig 8), water vapor competition effects in the observations are much more severe than in the model, indicating that observations are much more (positively) sensitive to updraft velocity. The opposite trends are seen for activation fraction ($\partial Nd/\partial Na$), given that reductions in aerosol reduce competition effects. The reduced water vapor competition effects at higher updraft velocities, and the trend in CCN error also generally explain why the sensitivities are smaller for the highest updraft velocity. '

At the end of the section we added the following concluding statement: ' Because of the discrepancy in sensitivities $\partial Nd/\partial Na$ and $\partial Nd/\partial w$, models may be predisposed to be too "aerosol-sensitive" or "aerosol insensitive" in aerosol-cloud climate interaction studies, even if they may capture average droplet numbers well. This is a subtle, but profound finding that only the sensitivities can clearly reveal and may explain inter-model biases on the aerosol indirect effect. Few published efforts (apart from Morales et al., 2014a and Sullivan et al., 2016) can demonstrate this, and non over a range of models and using a considerable aerosol dataset for evaluation as here performed.'

[Figure]

**Supplementary Figure S16.** Monthly ensembles for the years 2011-2015 of the CCN number concentration for supersaturation 0.2 % (CCN$_{0.2}$), 0.1% (CCN$_{0.1}$), 0.7% (CCN$_{0.7}$) and 1.0% (CCN$_{1.0}$) when observational data are available for Finokalia, Cabauw and Vavihill.

Specific comments

*p4 l12: Is "OA" really defined as a fraction here? - contradiction to p8 l8*

Reply: We thank the reviewer for pointing out this misleading sentence. The word 'fraction' is removed.

*p5 l13: really opposite, i.e. of different sign?*

Reply: We changed 'opposite' to 'different'

*p6 l30: Modal aerosol module*

Reply: corrected

*p7 l2: Kirkevåg*

Reply: corrected

*p7 l11: "a few models account for melting and sublimation of ice crystals" - I have the impression this sentence is incomplete.*

Reply: We added 'also'. The sentence now reads: 'In addition, while all models account for in-cloud scavenging of aerosols and for the aerosol release from evaporation droplets, a few models account also for melting and sublimation of ice crystals'

*p10 l15: why this very coarse resolution, and not just the resolution of the coarsest-resolved model?*

Reply: This is very close to the coarsest-resolved model of $4^o$x$5^o$ (as shown in the supplementary Table S1).

*p11 l6: the conclusion that the correlation is captured "satisfactorily" needs quantification: what can be considered "satisfactory"? How do the authors quantify these relative differences? - perhaps a figure like Fig. 3 but for the spatial variability would be useful?*

Reply: This part of the discussion refers to the CCN at 0.2% supersaturation. To address this issue, we have plotted the mean $CCN_{0.2}$ as calculated from the observations and as computed from the daily MMM for the days with available observations (figure here-below). In this figure the stations have been ranked based on $CCN_{0.2}$ observations in decreasing levels and this is nicely followed by the MMM with the exception of Finokalia station where the models overestimate $CCN_{0.2}$ as shown in Figure 2 and already discussed in the manuscript.
We also calculated the Pearson linear correlation coefficient (r) for the correlation of observed and modelled $CCN_{0.2}$ at each station as indicators of the ability of MMM to reproduce the temporal variability in the observations and found r values between 0.44 (for Melpitz) and 0.83 (for Mace Head), showing significant covariation of model results with observation.
We have modified the discussion accordingly:

'Despite the quantitative differences in the estimation of the $CCN_{0.2}$ concentrations, models are able to qualitatively capture the relative differences in $CCN_{0.2}$ concentrations between stations, as well as their seasonal variations. Comparing the $CCN_{0.2}$ as calculated from the observations and as computed from the daily MMM for the days with available observations for the stations, we find significant Pearson linear correlation coefficient (r) that vary between 0.44 (for Melpitz) and 0.83 (for Mace Head), showing significant correlation. Furthermore, ranking the stations based on the observed mean $CCN_{0.2}$ levels (supplementary figure S17) we find the corresponding MMM mean follows this station ranking with the exception of Finokalia where, as further discussed, the models overestimate the observed $CCN_{0.2}$ although they capture well (r=0.76) the observed temporal variability.'

[Figure]

**New supplementary figure S17**: Mean $CCN_{0.2}$ as calculated from the observations and as computed from the daily MMM for the days with available observations. The stations have been ranked based on $CCN_{0.2}$ observations in decreasing levels.

*p11 l7: The index of agreement, since it is not a conventional metric, would need to be defined in the main text. The authors further should motivate why this IoA provides extra information that substantially goes beyond NMB and NME.*

Reply: We provide such information at the end of section 2. 'For the comparison of model results with observations, a number of statistics variables have been calculated and defined as shown in the supplementary material S3.2. Hereafter we discuss

$$\text{the Index-of-Agreement } (IOA = 1 - \frac{\sum_{i=1}^{N}(P_i - O_i)^2}{\sum_{i=1}^{N}(|O_i - \bar{O}| + |P_i - \bar{O}|)^2}),$$

$$\text{the normalized mean bias } (NMB = \frac{\sum_{i=1}^{N}(P_i - O_i)}{\sum_{i=1}^{N}O_i} \times 100\%)$$

$$\text{and the normalized mean error } (NME = \frac{\sum_{i=1}^{N}|P_i - O_i|}{\sum_{i=1}^{N}O_i} \times 100\%),$$

where M are model results, O are observations and NMB, NME and IoA are used to quantify the performance of the models to reproduce observations. IoA is a measure of the agreement of model results with the observations. In this study we use all three for the evaluation of the capability of the models to reproduce the observations. We now calculate also

$$\text{the Pearson linear regression coefficient } (r = \left[\frac{\sum_{i=1}^{N}(P_i - \bar{P})(O_i - \bar{O})}{\sqrt{\sum_{i=1}^{N}(P_i - \bar{P})^2}\sqrt{\sum_{i=1}^{N}(O_i - \bar{O})^2}}\right])$$

as a measure of the ability of the models results to represent the variability in the observations.

*p13 l1: "satisfactorily" - again a quantification would be helpful. Is this significantly better than for CCN0.2?*

Reply: The performance of the model to simulate the observed seasonal variability can be quantified by the correlation coefficient. We have now calculated the statistics and the corresponding correlation coefficients for the MMM and we have modified the discussion accordingly to provide this information in the manuscript.

"Comparing model performance for CCN at low supersaturation ($CCN_{0.2}$, Figure 2) and at high supersaturation ($CCN_{1.0}$, Figure 4), $CCN_{1.0}$ is systematically underestimated by the models across all stations. The NME of MMM for $CCN_{0.2}$ ranges from 45% (Finokalia) to 81% (Jungfraujoch) for the different stations with significant correlation coefficients between 0.44 (Melpitz) and 0.86 (Mace Head) indicating that the model are able to simulate the temporal variability in the observations. For CCN at the highest supersaturation with available observations the NME varies from 50% (Finokalia) to 74% (Mace Head) and the correlation coefficients from 0.37 (Melpitz) to 0.78 (Mace Head) (see also supplementary Table S6). These results indicate that $CCN_{0.2}$ is in general better captured than CCN at higher supersaturations, both in absolute values and in temporal variability."

*p14 l5: the + for NME is unnecessary*

Reply: we will remove it

*p14 l10-12: Why would the authors call an NME of 97% "reasonable"?*

Reply: We rephrased this discussion p14-l8-20 of the ACPD version as follows:
"Figure S1 is similar to Figures 2 and 4 but shows particulate SO4, OA mass in $PM_1$ particles at the nine stations as well as model results for DU and SS.  Strong seasonal variations of the SO4 mass of about one order of magnitude are observed and simulated at the alpine site, Jungfraujoch and at the coastal background stations, Mace Head and Finokalia, although winter minima are overestimated by the models at these coastal sites. Smaller and no clear seasonal variation of SO4 is observed at the boreal forest environment of Hyytiälä, the rural background station Cabauw and at the highly polluted Melpitz station during the year. At these three stations, the MMM underestimates the observed annual mean concentration of SO4. Strong seasonal variations of the OA mass are observed and simulated at Mace Head, Finokalia, Jungfraujoch and Hyytiälä, while no distinct seasonal cycle in organic mass is seen at Cabauw and Melpitz and the OA concentrations are underestimated by MMM at all sites. The IoA between the MMM and the observations is between 0.28 and 0.62 for all stations. A detailed analysis of each model separately (Supplementary Table S6) shows that the OA mass concentration is underestimated (mean NMB is -37%) by nine of the models and overestimated by six of them (range of NMB -97% to 216%).

*p15 l6: I rather have the impression that it is completely unclear: in 5/9 station, it is indeed longer in winter, but in 4/9 the opposite.*

Reply: We agree with the reviewer. We added the scores. The sentence now reads:

'Depending on the station, the persistence time is longer during winter (5 stations) than during summer (4 stations).'

*p15 l7: in 6/9 cases, the MMM is consistent with the summer-winter change as the obs show, in 3/9 cases, opposite. Is that "qualitatively capturing" the change?*

Reply: We rephrase the sentence to be more precise as suggested by the reviewer:
'The average persistence of the $CCN_{0.2}$ number concentrations simulated by the individual models is consistent with the observed change between winter and summer at 6 among the 9 stations.'

*p15 l32: it doesn't hinder an eventual decrease in smax, it implies it.*

Reply: The sentence now reads:

'Increases in CCN concentrations tend to increased $N_d$ and associated water vapor depletion in the early stages of cloud formation; this in turn hinders the development of supersaturation and implies an eventual decrease in $s_{max}$.'

*p16 l24: The competition effect cannot explain why the ratio observed/simulated for Cabauw and Vavihill turns from substantial overestimation to underestimation, and why the opposite is found for Finokalia.*

Reply: See detailed reply to reviewer's major comments.

*p31 Fig. 2 – I don't see the dashed lines for the observations.*

Reply: Following reviewer's 1 comment we replaced 'dots and dashed lines' by 'symbols' and we have redrawn the figures to remove the dashed lines that were not clearly seen from the observations.

*p34 Fig. 5 – Would it be useful to show the MMM as well?*

Reply: Figure 5 has been redrawn to include MMM as suggested by the reviewer. The figure caption and the discussion have been modified accordingly.

References

Morales Betancourt, R. and Nenes, A.: Understanding the contributions of aerosol properties and parameterization discrepancies to droplet number variability in a global climate model, Atmos. Chem. Phys., 14(9), 4809–4826, doi:10.5194/acp-14-4809-2014, 2014a.

Morales Betancourt, R., and Nenes, A.: Aerosol Activation Parameterization: The population splitting concept revisited, Geosci.Mod.Dev., 7, 2345–2357, 2014b.

Sullivan, S.C., Lee, D., Oreopoulos, L., and Nenes, A (2016) The role of updraft velocity in temporal variability of cloud hydrometeor number, Proc.Nat.Acad.Sci., **113**, 21, 5781-5790, doi: 10.1073/pnas.1514043113